# Forest production efficiency increases with growth temperature

A. Collalti [1,2], A. Ibrom [3✉], A. Stockmarr[4], A. Cescatti[5], R. Alkama [5], M. Fernández-Martínez [6], G. Matteucci [7], S. Sitch [8], P. Friedlingstein [9], P. Ciais [10], D. S. Goll [11], J. E. M. S. Nabel [12], J. Pongratz[12,13], A. Arneth[14], V. Haverd[15] & I. C. Prentice[16,17,18]

Forest production efficiency (FPE) metric describes how efficiently the assimilated carbon is partitioned into plants organs (biomass production, BP) or—more generally—for the production of organic matter (net primary production, NPP). We present a global analysis of the relationship of FPE to stand-age and climate, based on a large compilation of data on gross primary production and either BP or NPP. FPE is important for both forest production and atmospheric carbon dioxide uptake. We find that FPE increases with absolute latitude, precipitation and (all else equal) with temperature. Earlier findings—FPE declining with age—are also supported by this analysis. However, the temperature effect is opposite to what would be expected based on the short-term physiological response of respiration rates to temperature, implying a top-down regulation of carbon loss, perhaps reflecting the higher carbon costs of nutrient acquisition in colder climates. Current ecosystem models do not reproduce this phenomenon. They consistently predict lower FPE in warmer climates, and are therefore likely to overestimate carbon losses in a warming climate.

[1] National Research Council of Italy, Institute for Agriculture and Forestry Systems in the Mediterranean (ISAFOM), 06128 Perugia (PG), Italy. [2] University of Tuscia, Department of Innovation in Biological, Agro-food and Forest Systems (DIBAF), 01100 Viterbo, Italy. [3] Technical University of Denmark (DTU), Department of Environmental Engineering, Lyngby, Denmark. [4] Technical University of Denmark (DTU), Department of Applied Mathematics and Computer Science, Lyngby, Denmark. [5] European Commission, Joint Research Centre, Directorate for Sustainable Resources, Ispra, Italy. [6] Research group PLECO (Plants and Ecosystems), Department of Biology, University of Antwerp, 2610 Wilrijk, Belgium. [7] National Research Council of Italy, Institute for BioEconomy (IBE), 50019 Sesto Fiorentino, FI, Italy. [8] College of Life and Environmental Sciences, University of Exeter, Exeter EX4 4RJ, UK. [9] College of Engineering, Mathematics and Physical Sciences, University of Exeter, Exeter EX4 4QF, UK. [10] Laboratoire des Sciences du Climat et del'Environnement, CEA CNRS UVSQ, Gif-sur-Yvette 91191, France. [11] Department of Geography, University of Augsburg, Augsburg, Germany. [12] Max Planck Institute for Meteorology, Hamburg, Germany. [13] Ludwig-Maximilians-Universität München, Luisenstr 37, 80333 Munich, Germany. [14] Karlsruhe Institute of Technology, Institute of Meteorology and Climate Research/Atmospheric Environmental Research, 82467 Garmisch-Partenkirchen, Germany. [15] CSIRO Oceans and Atmosphere, Canberra, ACT 2601, Australia. [16] Department of Life Sciences, Imperial College London, Silwood Park Campus, London Ascot SL5 7PY, UK. [17] Department of Biological Sciences, Macquarie University, North Ryde, NSW 2109, Australia. [18] Department of Earth System Science, Tsinghua University, 100084 Beijing, China. ✉email: anib@env.dtu.dk

A utotrophic respiration releases to the atmosphere about half (~60 PgC yr$^{-1}$) of the carbon fixed annually by photosynthesis[1]. Forest ecosystems are the largest carbon sink on land, taking up about $3.5 \pm 1.0$ PgC yr$^{-1}$ (2008–2017) on average[2]. A small change in the proportion of carbon losses, for example due to climate change, would strongly affect the net carbon balance of the biosphere. Predicting the autotrophic component of the carbon balance of forests under changing climate requires understanding of how much atmospheric $CO_2$ is assimilated through photosynthesis (gross primary production, GPP), how much is released due to plant metabolism (autotrophic respiration, $R_a$), how efficiently plants use assimilated carbon for the production of organic matter (net primary production, NPP), and how organic carbon is partitioned into plant organs (biomass production, BP) versus other less stable forms—which include soluble organic compounds exuded to the rhizosphere or stored as reserves, and biogenic volatile organic compounds (BVOCs) emitted to the atmosphere[3].

The climate sensitivity of the terrestrial carbon cycle can be benchmarked using ratios between these fluxes across a range of climates. We focus here on the ratio of NPP to GPP, the so called carbon use efficiency (CUE = NPP/GPP) and of BP to GPP, called biomass production efficiency (BPE = BP/GPP). The two concepts are close, but not identical[4,5]. BPE is substantially easier to obtain, because the additional fluxes that constitute NPP are notoriously difficult to measure. For this reason, there are far more data available on BPE, while uncertainties associated with both BP and NPP measurement make it impossible to distinguish them in large data compilations. Therefore, we assessed estimates of both BPE and CUE as a single metric, hereafter called forest production efficiency (FPE), but making distinctions between them when needed and when possible.

Over 20 years ago, the debate about spatial gradients of forest CUE seemed to be resolved by Waring et al.[6], who found CUE to be nearly constant ($0.47 \pm 0.04$: here and elsewhere, $\pm$ denotes one standard deviation) across temperate and boreal forest stands ($n = 12$). The assumption of a universal value for CUE—implying a tight coupling of whole-plant respiration to photosynthesis—has obvious practical convenience, and numerous vegetation models have adopted it[5]. Many complex process-based vegetation models, however, assume decoupling of photosynthesis and respiration, with the latter driven by temperature[7] and biomass[8]—implying that CUE must vary with changing environmental conditions. There is no general, observationally based consensus as to which of these two (mutually incompatible) model assumptions is nearer to the truth. One study found that BPE is greater at higher soil fertility[4], perhaps because less carbon needs to be allocated for nutrient acquisition. Forest management[9], stand age[10] and climate[11,12] have also been reported to influence CUE and BPE.

Here we revisit the global patterns of forest CUE and BPE considering multiple controls and the potential effects of methodological uncertainty, based on a large global set of data on forest CUE and/or BPE ($n = 244$), spanning environments ranging from the tropical lowlands to high latitudes and high altitudes (Supplementary Fig. 1).

Overall, we find that FPE decreases with age and increases with site factors such as annual air temperature, total annual precipitation and absolute latitude.

## Results

**FPE is not a universal constant.** Results show that both CUE ($0.47 \pm 0.13$; range 0.24–0.71; $n = 47$) and BPE ($0.46 \pm 0.12$; range 0.22–0.79; $n = 197$) have large variability; therefore, neither can be assumed to be uniform (Fig. 1 and Supplementary Fig. 2). CUE

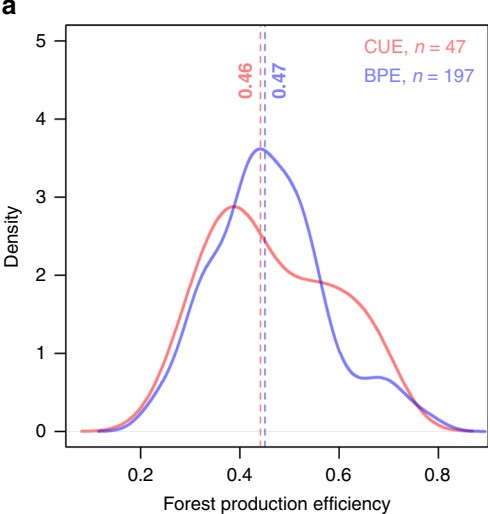

**Fig. 1 Carbon use efficiency vs. biomass production efficiency.** Density plot of carbon use efficiency (CUE, red line, $n = 47$) and biomass production efficiency (BPE, blue line, $n = 197$) data from all available data. The vertical lines are medians.

and BPE are statistically indistinguishable in our dataset because of uncertainties associated with both quantities ($\pm 0.39$ for CUE and $\pm 0.16$ for BPE: see Methods). Overall, the average FPE in our dataset ($0.46 \pm 0.12$; range 0.22–0.79; $n = 244$) is statistically indistinguishable from that provided by Waring et al.[6], but its standard deviation is three times larger (Methods and ref. [5]). Different GPP estimation methods produced slightly different distributions (Fig. 2a), with median values ranging from 0.42 (scaling; upscaling of chamber-based measurements) through 0.48 (micrometeorological; ecosystem-scale $CO_2$ flux measurements) to 0.48 (model; process-based models) (see Methods for definitions). Stand age had a further effect on FPE, as shown by the differing median CUE and BPE values of stands in intermediate (in the forestry sense, i.e. 20–60 years) and younger age classes, with FPE varying from 0.52 (age class <20 years) to 0.42 (age class >60 years) (Fig. 2b). Figure 3 shows how the data compare to those published by Waring et al.[6]. The small variability of CUE reported by Waring et al.[6] was already noted by Medlyn & Dewar[13] as untypical, and artificially constrained by the method used to calculate CUE. Medlyn & Dewar[13] suggested a 0.31–0.59 range as being realistic. Figure 3 also indicates systematically lower CUE than Waring et al.[6] for forests with GPP < ~2000 gC m$^{-2}$ yr$^{-1}$, especially in forests in the old age class; and a tendency to higher values for forests with GPP > ~2000 gC m$^{-2}$ yr$^{-1}$ and in the young age class.

**Factors controlling FPE variability.** We used mixed-effects multiple linear regression to infer the multiple drivers of the spatial pattern of FPE. This method separates the contribution of every predictor variable included in the analysis, even if they are correlated to some degree (Methods). Four predictors—out of an initial selection of eleven (listed in Methods)—proved to be important: stand age (age, years), mean annual temperature (MAT, °C), total annual precipitation (TAP, mm year$^{-1}$) and absolute latitude (|lat|, °), all included as fixed effects (Fig. 4). The method used to measure GPP (GPP method)—was included as a random effect (Table 1, Eq. (1)).

The use of multiple regression was essential for this analysis. Simple correlations between FPE and individual predictors showed no significant effects, while there were significant correlations among the predictors (Supplementary Table 1).

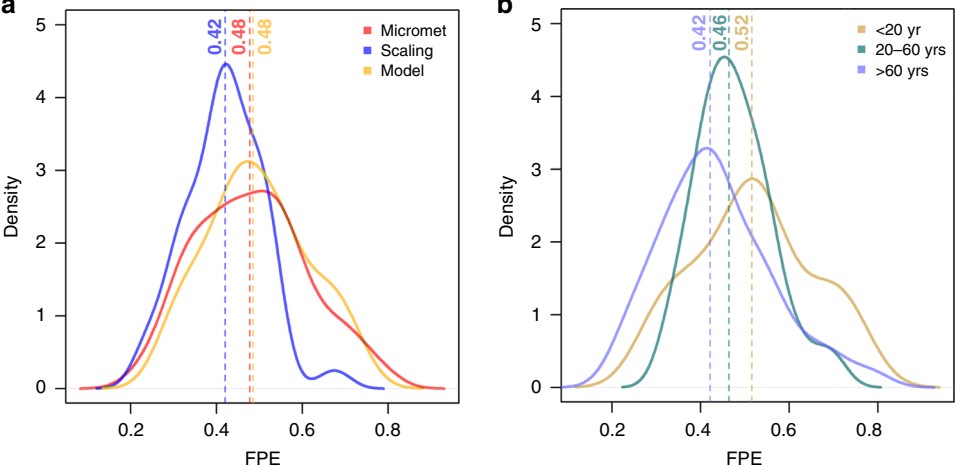

**Fig. 2 Effects of GPP method and age on FPE variability. a** Forest production efficiency (FPE) density plots for three subsets of data where the GPP was estimated with three different methods (micrometeorological, red line, $n = 98$; scaling, blue line, $n = 73$; and models, green line, $n = 53$). The vertical lines are medians. **b** Density plots for different age classes (age < 20 years, light brown line, $n = 47$; 20–60 years, green line, $n = 49$; and age > 60 years, blue line, $n = 77$).

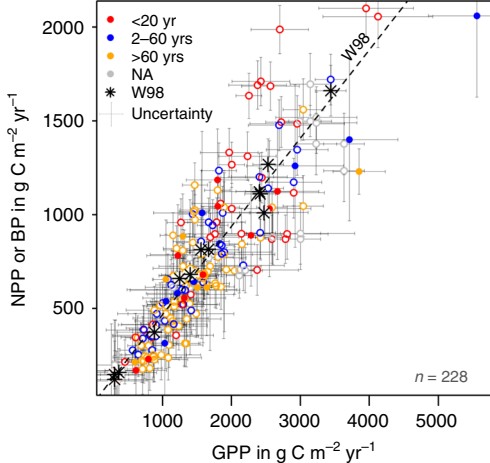

**Fig. 3 Comparison of the present work dataset vs. Waring et al.[6].** Scatter plot of net primary production (NPP, gC m$^{-2}$ yr$^{-1}$) or biomass production (BP, gC m$^{-2}$ yr$^{-1}$) versus gross primary production (GPP, gC m$^{-2}$ yr$^{-1}$) ($n = 228$). Open circles: BP, filled circles: NPP. Stars represent data points from Waring et al.[6]. The line marked with W98 represents a CUE (i.e. NPP/GPP) of 0.47. Age classes are marked by colours (see top left of the figure); NA stands for 'age not available'. The uncertainty (gC m$^{-2}$ yr$^{-1}$) of the data points is indicated by bars (for data uncertainty see Methods).

The final model with four fixed effects is:

$$\text{FPE} = \beta_0 + \beta_1 \text{MAT} + \beta_2 \text{age} + \beta_3 \text{TAP} + \beta_4 |\text{lat}| + \eta Z_{\text{GPPMeth}} + \varepsilon \quad (1)$$

where $\beta_0$ is the intercept, $\beta_1$–$\beta_4$ are the estimated sensitivities of FPE to MAT, stand age, TAP and $|\text{lat}|$; $\eta Z_{\text{GPPMeth}}$ is a random intercept for two distinguishable 'GPPmethod' classes and $\varepsilon$ is the residual (Table 1). The model could not be further reduced at a 5% test level, i.e. omitting any one of these predictors yielded a significantly different model. Supplementary Table 2 lists the combinations that were tried.

We examined random effects from three methods to estimate GPP (Methods). The methods biometric and scaling constituted a single class, while GPP values determined from micrometeorological measurements were systematically higher (thus FPE was systematically lower). The multiple regression model explained 30% of the variance in the observed FPE values. Given the large uncertainty in the estimation of NPP and GPP values, and the structural and physiological diversity of the forests, this value was unexpectedly high.

We could not fit an independent statistical model for CUE, because there were too few sites with NPP ($n = 31$) measurements. Furthermore, adding a random intercept for the two categories (CUE or BPE) to Eq. (1) yielded almost identical values, of 0.47 for CUE and 0.46 for BPE.

We also applied the mixed-effects multiple regression model to the TRENDY v.7 outputs of eight Dynamic Global Vegetation Models (DGVMs) to examine whether the multivariate relationships shown for the FPE data could also be seen in the model simulations, in order to test whether the observed pattern would also emerge from the processes representations in the models. We originally aimed to use the same mixed-effects linear model, at the locations of the data points to fit the simulated FPE. However, because these DGVMs do not consider forest age, we had to alter the model equation to:

$$\text{FPE} = \mu_0 + \mu_1 \text{MAT} + \mu_2 \text{TAP} + \mu_3 |\text{lat}| + \eta Z_{\text{Model}} + \varepsilon \quad (2)$$

Neither this model, nor any model that could be derived from it, fulfilled the conditions of normally distributed residuals. In other words, the simulations did not represent a common emergent relationship consistent with the data. The most likely explanation is that the models use different parameters (and even sometimes different functional relationships) for different biomes, so that no general relationship applying across all forest types can be expected to emerge. This phenomenon is evident from Fig. 5 where many models show discontinuities in CUE.

CUE outputs from the TRENDY v.7 model ensemble, produced by the eight DGVMs, consistently showed a negative relationship with MAT—opposite to that shown by our analysis. The slope ($\partial$CUE/$\partial$MAT) estimated from data was +0.006 °C$^{-1}$ (see Table 1); the slopes from models ranged from –0.0025 °C$^{-1}$ for LPJ-GUESS to –0.0098 °C$^{-1}$ for SDGVM (Fig. 5). The average slope across the eight models was –0.005 °C$^{-1}$. All models showed high CUE for boreal forests and low CUE for tropical forests, but with considerable variation among models (Supplementary Fig. 4). The modelled CUE values agree well with the data only in temperate regions (MAT 5–15 °C, $n = 156$), but differ greatly in boreal (MAT < 5 °C, $n = 35$) and tropical (MAT > 15 °C, $n = 40$) regions.

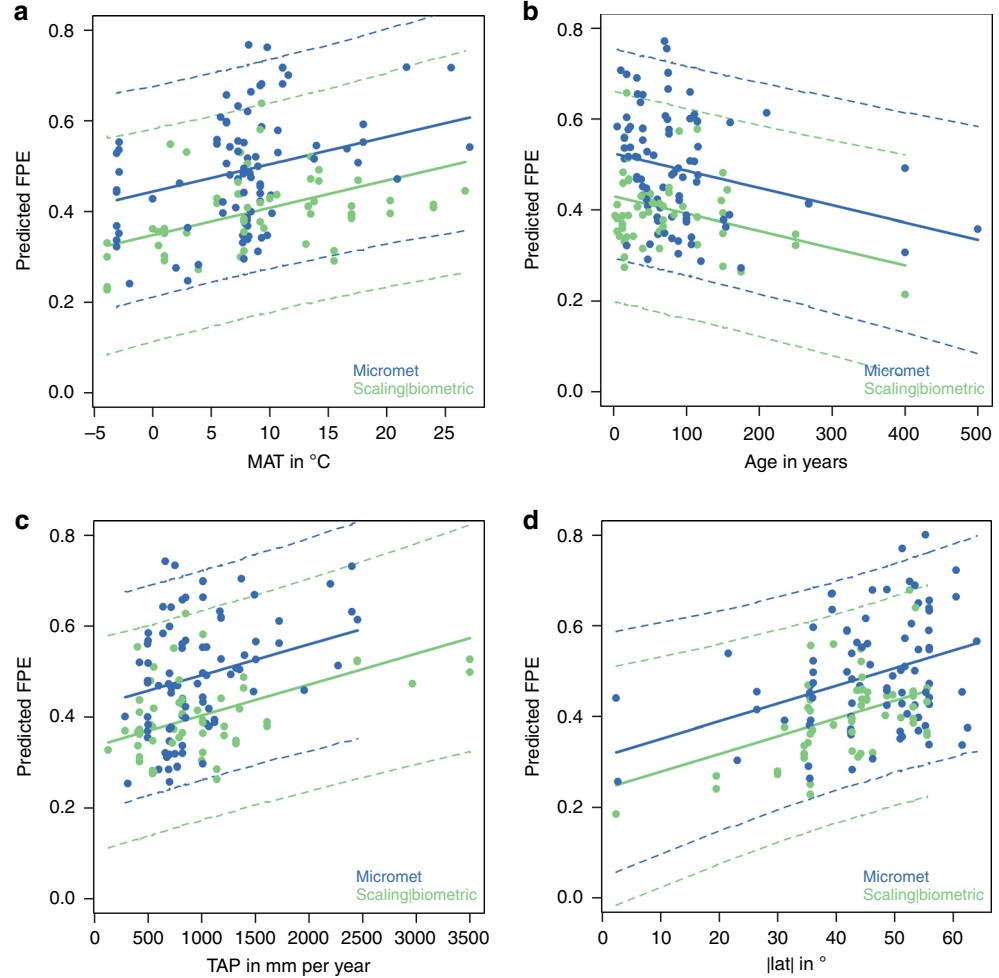

**Fig. 4 Predicted FPE vs. single effects of environmental and structural variables.** Predictions of the mixed linear model for single fixed effects (Eq. (1)), given the other independent variables constant at their average values for that GPP method category. The dashed lines represent confidence intervals at the 0.05 and 0.95 levels calculated with the function 'predict Interval' of the R-package 'merTools'.

**Table 1 Parameters of the mixed-effects multiple regression model (Eq. (1)).**

|  | Estimate | Std Error | df | t value | p value | Significance |
|---|---|---|---|---|---|---|
| Random intercepts |  |  |  |  |  |  |
| 'Micromet' | 0.23 |  |  |  |  |  |
| 'Scaling \| biometric' | 0.14 |  |  |  |  |  |
| Intercept ($\beta_0$) Slopes: | 0.19 | 0.106 | 24 | 1.77 | 0.09 | n.s. |
| MAT ($\beta_1$) | 0.0060 | 0.0025 | 136 | 2.45 | 0.016 | * |
| age ($\beta_2$) | −0.00038 | 0.000116 | 136 | −3.28 | 0.0013 | ** |
| TAP ($\beta_3$) | 6.8E-5 | 2.07E-05 | 136 | 3.28 | 0.0014 | ** |
| \|lat\| ($\beta_4$) | 0.0039 | 0.0016 | 136 | 2.45 | 0.016 | * |

Parameter estimate of coefficients in Eq. (1) and their standard errors (Std. Error), degrees of freedoms (df), t- and p values of the two-sided *t*-test and the ANOVA (*$p < 0.05$, **$p < 0.01$, ***$p < 0.001$). The squared Pearson's correlation coefficient and the squared Spearman's correlation values are both equal to 0.31.
*MAT* mean annual temperature, *age* stand age, *TAP* total annual precipitation, *|lat|* absolute latitude.

## Discussion

**The empirical ranges of CUE and BPE and age effects on FPE.** Under some extreme circumstances, the carbon flux to mycorrhizae and root exudates can constitute as much as 50% of daily assimilation[14] or as much as 30% of annual NPP[15,16]. While in nonstressed conditions BVOCs consume a small fraction (~5% or less) of annual NPP, under stressed conditions and in hot climates, BVOC emissions can consume 15–50% of annual NPP[17,18]. Thus CUE– if not equal to BPE –, should always be larger than BPE. In our dataset, in those cases where both could

be estimated, CUE was larger than the estimated BPE in seven out of thirteen cases (Supplementary Information, Supplementary Table 3). In the remaining cases, the estimated CUE was statistically indistinguishable from BPE. This finding suggests that the fraction of these unaccounted organic carbon flows varies substantially among forests.

Statistically fitted values of BPE and CUE ranged between 0.27 (−0.04) and 0.58 (+0.04). The numbers in parentheses for CUE and BPE reflect our estimates of methodological bias (random intercepts, see Table 1). Ninety-two percent of BPE and CUE

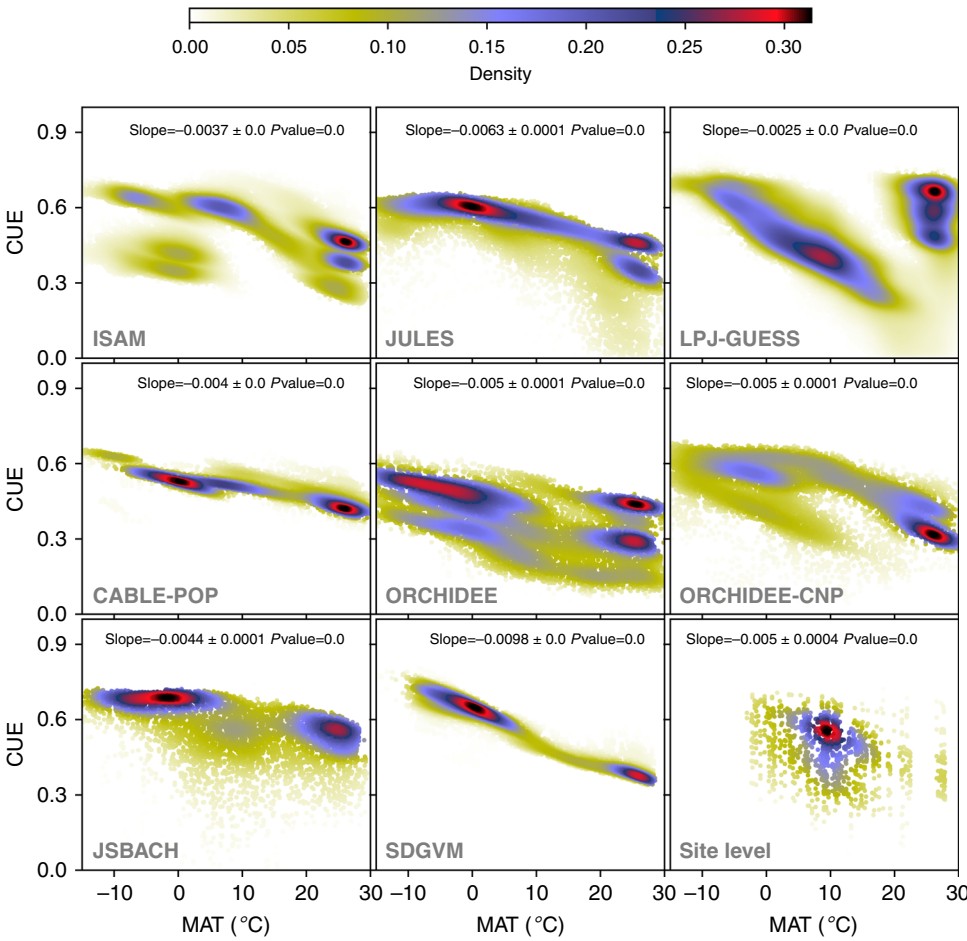

**Fig. 5 Modelled TRENDY v.7 CUE and growth temperature patterns.** Density plots (i.e. frequency of forest carbon use efficiency (CUE) value divided by the total number of grid cells of simulated CUE derived from the following TRENDY v.7 process-based models: ISAM, JULES, LPJ-GUESS, CABLE-POP, ORCHIDEE, ORCHIDEE-CNP, JSBACH and SDGVM, averaged from 1995 to 2015 as function of MAT (°C). In the last (right bottom) density plot, data points extracted from coordinates and times of observed sites and used to plot the simulated CUE as function of MAT from the eight TRENDY v.7 models.

values in the dataset lie within the allowable range according to Amthor[19], reflecting maximum growth with minimum expenditure (0.65) and minimum growth with maximum maintenance costs (0.2). The remaining 8% of the data exceed the upper bound given in ref. [19]. However, no values below 0.2 were found, and it seems likely such values cannot be physiologically sustained by plants for long periods[8]. They might be encountered in moribund stands, unlikely to be sampled (perhaps an example of survivorship bias, or the desk drawer problem[20]). However, values >0.65 apparently can (temporarily) occur in young, actively growing forests.

Age- or size-related declines in both GPP, NPP and FPE (as CUE or BPE) have been reported in earlier studies[9,11,21,22] but a decline in FPE is shown unequivocally here (slope $\partial$FPE/$\partial$age = −0.0004 yr$^{-1}$), based on a larger dataset than previously analysed[23–25] (Fig. 4 and Table 1). This decline could have several contributory causes. First, the longer transport pathway for water in taller trees can result in more closed stomata (to avoid xylem cavitation) and therefore reduced GPP[26], with no corresponding reduction in $R_a$, at least in the short-term[27]. Second, larger trees may respire more because of their greater sapwood volume and mass per unit leaf area[8,28,29], leading to increased $R_a$ (for the maintenance of living sapwood tissues) and reduced NPP relative to GPP. Third, soil fertility declines due to nutrient immobilization as stands age[30]; this is consistent with observations of an increased ratio of fine-root-to-leaf-carbon, and

reduced nitrogen concentration in soils[10,31]. Ontogenetic shifts from structural biomass to reserve allocation, and structural and resource limitations in older stands, are also all expected to decrease production efficiency[32]. Conversely, reducing plant competition and rejuvenating stands through forest management should tend to increase both CUE and BPE[9,33]. At the stand scale, closing canopies may contribute further to reducing or stabilizing the GPP[34] of individual trees. It is also likely that young trees allocate more carbon to biomass growth as they compete for light and nutrients; while older trees invest more in maintenance of their existing biomass, and prioritize the chemical defence of that biomass, relative to acquisition of new biomass[35,36].

An additional hypothesis[37] invokes an increase in nonstructural carbohydrates (NSC) allocation as trees grow. NSC is a substantial carbon pool, containing in some cases up to four times the carbon content of leaves in the canopy and increasing as trees increase in size[38]. An increased flux to NSC however would imply a reduced BPE, but not a reduced CUE.

**Environmental effects on FPE.** The increase in FPE with increasing annual precipitation, to our knowledge, has not been noted previously. Higher TAP results in increasing soil water availability and greater stomatal openness, which might imply increased photosynthesis. There is no direct evidence that water availability influences autotrophic respiration; on the other hand, respiration has been found to increase with drought[39].

With increasing TAP, photosynthesis might be expected to increase faster than plant respiration, leading to higher CUE and BPE.

The increase of FPE with absolute latitude has also, to our knowledge, not been described before. Higher latitudes experience longer days in summer. The diffuse fraction also increases with the path-length of radiation through the atmosphere. Both effects might be expected to increase the radiation use efficiency of GPP. While high irradiances in the tropics leads to saturation of photosynthesis in the uppermost leaf layers[40], they also allow for higher leaf area to utilize the transmitted radiation in the relatively short daylight hours. Higher leaf area also implies higher $R_a$ and lower FPE. The latitude effect compensates for the MAT effect, because these two variables are negatively correlated. Despite the correlation, the linear mixed model is able to distinguish between the individual effects of MAT and |lat|. This can be demonstrated by comparison of high-latitude with high-elevation sites at the same MAT. The effects of radiation are incorporated in vegetation models which, therefore, could in principle represent radiation regime effects on FPE.

Moreover, as far as we know, the observed increase in FPE with mean annual temperature has not previously reported. This increase is opposite to what would be expected based on the instantaneous responses of photosynthesis and plant maintenance respiration as described in textbooks[41] and assumed in many process-based models (Fig. 5). The instantaneous response of maintenance respiration to a temperature change is steeper than that of photosynthesis[42]. Moreover, under natural conditions photosynthesis is commonly limited by light, while respiration is not. However, the instantaneous response of autotrophic respiration rate is largely irrelevant here because of the longer time scale. A long line of investigations, starting with Gifford[43], has shown the ubiquity of respiratory thermal acclimation, whereby the effect of increased growth temperature on enzyme kinetics is offset by a lowering of the base rate[44]. This acclimation takes place on a time scale of days to weeks[1]. Genetic adaptation throughout multiple generations is expected to proceed in the same direction (for definitions and distinctions between acclimation and adaptation see ref. [45]). One consequence of these processes is that observed rates of maintenance respiration vary with temperature (in both space and time) far less steeply than would be expected based on the instantaneous response of enzyme kinetics[46]. This has been shown comprehensively in leaves, and is likely to apply to all plant tissues[1]. Moreover, the ratio of respiration to carboxylation capacity, assessed at growth temperature, is slightly but significantly larger in colder climates[46].

He et al.[47] found—in contrast to our results—a latitudinal pattern with higher CUE at high latitudes declining nonlinearly with increasing MAT and stabilizing at increasing TAP. These results were obtained using an 'emergent constraint' method to narrow the range of global mean carbon use efficiency values produced by an ensemble of ecosystem models. The observed correlation between simulated global and site-specific CUE was used to translate the probability distribution of observed site CUE into a distribution of global CUE. This method's validity, however, depends on the models correctly representing the relationship between site-specific and global CUE. Thus, the findings of ref. [47] could simply reflect the standard assumption of models that $R_a$ increases with temperature more steeply than GPP[42]. We have shown the same patterns here in all of the TRENDY v.7 ensemble simulations (Fig. 5) but our analysis shows that the underlying assumption is incorrect.

Adaptive mechanisms, potentially contributing to respiratory thermal acclimation, include changes in the physiology and growth of active tissues (i.e. the relation between assimilating and non-assimilating tissues) and changes in the amount of enzymes and their activation states to match substrate availability[42,48]. Heat tolerance in leaves has also been found to increase linearly with temperature and to decrease with absolute latitude[49]. Therefore, a simple explanation for the increase of FPE with temperature might be that plants can achieve the same function at a higher temperature with smaller amounts of enzymes, thereby decreasing the respiratory losses incurred during the maintenance of catalytic capacity. Especially low FPE in boreal forests could be the consequence of greater allocation of assimilates to nutrient acquisition (via root exudation and exports to mycorrhizae) in cold soils where microbial activity is much lower than in tropical forests[31,50]. Low FPE in cold climates may also reflect the need to repair tissues affected by frost damage[51].

**Whole-plant constraints and consequences for modelling.** Amthor[19] derived an upper bound of 0.65 for CUE, based on a rough quantification of the minimum respiratory costs for plants to function. His lower bound of 0.2 was based on the need for a sufficiently positive carbon balance to have minimum photosynthesis to survive and to allow trees to compensate for tissue turnover, reproduction and mortality. However, most CUE values lie within narrower bounds, suggesting the existence of additional regulatory mechanisms at the whole-plant scale. Gifford[43] noted that autotrophic respiration and primary production are interdependent, because carbon must be assimilated before it is respired, while respiration is required for the growth and maintenance of tissues. He opined that: 'Plant respiratory regulation is too complex for a mechanistic representation in current terrestrial productivity models for carbon accounting and global change research' and indicated a preference for simpler approaches that capture the essence of the process. The opposite view was expressed by Thornley[52], who argued that: 'attempting to grasp and pin down complexity is often the first step to finding a way through a labyrinth'. Without taking a position on this controversy, we note that the standard approach in most of today's land ecosystem models, or more generally in vegetation models—where maintenance respiration per unit of respiring tissue is typically determined as a fixed basal rate at a standard temperature (commonly 15 or 20 C°), increasing with the substrate and temperature according to a fixed $Q_{10}$ factor or Arrhenius-type equation—cannot generate the positive response of CUE or BPE to growth temperature observed in our study. Moreover, as shown in Fig. 5, the presence of discontinuities in CUE probably represents an attempt to sidestep an inevitable consequence of this incorrect approach. Unless plant functional types from warmer environments are assigned lower basal maintenance respiration rates, modelled CUE becomes implausibly low in warm climates. However, the idea of assigning fixed basal maintenance respiration rates to plant types has no observational or experimental basis.

In contrast, the use of production efficiency concepts in models seems well motivated[53], provided they are not assumed to be constant across different stands and environments. Production efficiency is a valuable unifying concept for the analysis of forest carbon budgets. Although more variable than was once thought, FPE appears to be a relatively conservative quantity, subject to inherent biological constraints, that has demonstrable relationships to stand development, latitude and climate. The possible explanations for the observed global multifactorial pattern in FPE give rise to hypotheses on how vegetation models might incorporate whole-plant regulation mechanisms of the carbon losses for a given stand. The demonstrated empirical pattern should then be used to constrain new model developments.

## Methods

**Definitions of terms.** GPP is defined here as 'the sum of gross carbon fixation (carboxylation minus photorespiration) by autotrophic carbon-fixing tissues per unit area and time[54]. GPP is expressed as mass of organic carbon produced per unit area and time, over at least one year. NPP consists of all organic carbon that is fixed, but not respired over a given time period[54]:

$$NPP = GPP - R_a = \Delta B + L + F + H + O = BP + O \quad (3)$$

with all terms expressed in unit of mass of carbon per unit area and time. $R_a$ is autotrophic respiration (composed of growth and maintenance respiration components); $\Delta B$ is the annual change in standing biomass carbon; litter production (roots, leaves and woody debris) is $L$; fruit production is $F$; the loss to herbivores is $H$, which was not accounted here because of the very limited number of observations available. BP is biomass production[4]. Symbol $O$ represents occult, carbon flows, i.e. all other allocations of assimilated carbon, including changes in the nonstructural carbohydrate pool, root exudates, carbon subsidies to symbiotic fungi (mycorrhizae) or bacteria (e.g. nitrogen fixers), and BVOCs emissions (Supplementary Fig. 1). These 'occult' components are often ignored or unaccounted when estimating NPP, hence this bias is necessarily propagated into the $R_a$ estimate when $R_a$ is calculated as the difference between GPP and NPP[55].

**Estimation methods.** We grouped the 'methods' into four categories:

- biometric: direct tree stock measurements, or proxy data together with biomass expansion factors, allometric equations and the stock change as a BP component. If not otherwise stated, we assumed that the values included both above- and below-ground plant parts ($n = 13$ for GPP; $n = 200$ for NPP or BP).
- micrometeorological: micrometeorological flux measurements using the eddy-covariance technique to measure $CO_2$ flux and partitioning methods to estimate ecosystem respiration and GPP ($n = 98$ for GPP; $n = 4$ for NPP or BP).
- model: model applications ranging from single mathematical equations (for canopy photosynthesis and whole-tree respiration) to more complex mechanistic process-based models to estimate GPP and $R_a$, with NPP as the net difference between them ($n = 53$ for GPP; $n = 24$ for NPP or BP).
- scaling: upscaling of chamber-based measurements of assimilation and respiration (GPP and $R_a$) fluxes at the organ scale, or the entire stand ($n = 73$ for GPP; $n = 9$ for NPP or BP).

The difference between 'scaling' and 'modelling' lies in the data used. In the case of 'scaling' the data were derived from measurements at the site. 'Model' means that a dynamic process-based model was used, but with parameters calibrated and optimized at the site, based on either biometric or micrometeorological measurements.

**Data selection.** The data were obtained from more than 300 peer-reviewed articles (see also ref. [5]), adding, merging and extending published works worldwide on CUE or BPE[4,9,11,23,25,56,57]. Data were extracted from the text, Tables or directly from Figures using the Unix software g3data (version 1.5.2, Jonas Frantz). In most studies, NPP, BP and GPP were estimated for the tree stand only. However, GPP estimated from $CO_2$ flux by micrometeorological methods applies to the entire stand including ground vegetation. We therefore included only those micrometeorological studies where the forest stand was the dominant primary producer. The database contains 244 records (197 for BPE and 47 for CUE) from >100 forest sites (including planted, managed, recently burned, N-fertilized, irrigated and artificially $CO_2$-fertilized forests; Supplementary Information, Supplementary Fig. 3 and online Materials; https://doi.org/10.5281/zenodo.3953478), representing 89 different tree species. Globally, 170 records out of the total data are from temperate sites, 51 from boreal, and 23 for tropical sites, corresponding to 79 deciduous broad-leaf (DBF), 14 evergreen broad-leaf (EBF), 132 evergreen needle-leaf (ENF) and 19 mixed-forests records (MX). The majority of the data (∼93%) cover the time-span from 1995 to 2015. We assume that when productivity data came from biometric measurements the reported NPP would have to be considered as BP because 'occult', nonstructural and secondary carbon compounds (e.g. BVOCs or exudates) are not included. In some cases, multiple datasets from the same site were included, covering different years or published by different authors. We considered only those values where either NPP (or BP) and GPP referred to the same year. From studies where data were available from more than 1 year, mean values across years were calculated. When the same reference for data was found in different papers or collected in different databases, where possible, we used data from the original source. When different authors described the same values for the same site, one single reference (and value) was used (in principle the oldest one). By using only commonly available environmental drivers to analyse the spatial variability in CUE and BPE, we were able to include almost all of the data that we found in the literature. We examined as potential predictors site-level effects of: average stand age ($n = 204$; range from 5 to ∼500 years), mean annual temperature (MAT; $n = 230$; range −6.5 to 27.1 °C) and total annual precipitation (TAP; $n = 232$; range from ∼125 to ∼3500 mm yr$^{-1}$), method of determination ($n = 237$), geographic location (latitude and longitude; $n = 241$, 64°07′N to −42°52′S and 155° 70′W to −173°28′E), elevation ($n = 217$; 5–2800 m, above sea level), leaf area index

(LAI; $n = 117$; range from 0.4 to 13 m$^2$ m$^{-2}$), treatment (e.g.: ambient or artificially increased atmospheric $CO_2$ concentration; $n = 34$), disturbance type (e.g.: fire $n = 6$; management $n = 55$), and the International Geosphere-Biosphere Programme (IGBP) vegetation classification and biomes ($n = 244$), as reported in the published articles (online Materials). The methods by which GPP, NPP, BP (and $R_a$) were determined were included as random effects in a number of possible mixed-effects linear regression models (Supplementary Table 4).

We excluded from statistical analysis all data where GPP and NPP were determined based on assumptions (e.g. data obtained using fixed fractions of NPP or $R_a$ of GPP). In just one case GPP was estimated as the sum of upscaled $R_a$ and NPP[58]; however, this study was excluded from the statistical analysis. NPP or $R_a$ estimates obtained by process-based models ($n = 23$) were also not included in the statistical analysis. No information was available on prior natural disturbance events (biotic and abiotic, e.g. insect herbivore and pathogen outbreaks, and drought) that could in principle modify production efficiency, apart from fire. The occurrence of fire was reported by only a few studies[59–61]. These data were included in the database but fire, as an explanatory factor, was not considered due to the small number of samples in which it was reported ($n = 6$).

**Data uncertainty.** Uncertainties of GPP, NPP and BP data were all computed following the method based on expert judgment as described in Luyssaert et al.[55]. First, 'gross' uncertainty in GPP (gC m$^{-2}$ yr$^{-1}$) was calculated as $500 + 7.1 \times (70 - |$lat$|$) gC m$^{-2}$ yr$^{-1}$ and gross uncertainties in NPP and BP (gC m$^{-2}$ yr$^{-1}$) were calculated as $350 + 2.9 \times (70 - |$lat$|$). The absolute value of uncertainty thus decreases linearly with increasing latitude for GPP and for NPP and BP, because we assumed that the uncertainty is relative to the magnitude of the flux, which also decreases with increasing |lat|. Subsequently, as in Luyssaert et al.[55], uncertainty was further reduced considering the methodology used to obtain each variable, by a method-specific factor (from 0 to 1, final uncertainty ($\delta$) = gross uncertainty × method-specific factor). Luyssaert et al.[55] reported for GPP-Micromet a method-specific factor of 0.3 (i.e. gross uncertainty is reduced by 70% for micrometeorological measurements); and for GPP-Model, 0.6. GPP-Scaling and GPP-Biometric were not explicitly considered in ref. [55] for GPP. We we used values of 0.8 and 0.3, respectively. For BP-Biometric and NPP-Micromet we used a reduction factor of 0.3; for NPP-Model, 0.6; and for NPP-Scaling (as obtained from chamber-based $R_a$ measurements), 0.8. When GPP and/or NPP or BP methods were not known ($n = 7$), a factor of 1 (i.e. no reduction of uncertainty for methods used, hence maximum uncertainty) was used. The absolute uncertainties on CUE ($\delta$CUE) and BPE ($\delta$BPE) were considered as the weighted means[62] by error propagation of each single variable ($\delta$NPP or $\delta$BP and $\delta$GPP) as follows:

$$\delta CUE = \sqrt{\left(\frac{\delta NPP}{GPP}\right)^2 + \left(\delta GPP \frac{NPP}{GPP^2}\right)^2} \quad (4)$$

and similarly for $\delta$BPE, by substituting NPP with BP and CUE with BPE.

**Data and model selection.** The CUE and BPE data were combined into a single variable, as sites for which both types of estimates existed did not show any significant differences between these entities (Supplementary Fig. 2). CUE values based on modelling were excluded (in our database we do not have BPE data from modelling). Tests showed that the CUE value was systematically higher when GPP was estimated with micrometeorological methods, compared to values based on biometric or scaling methods. Only data with complete information on CUE, MAT, age, TAP, and latitude were used. Altogether, 142 observations were selected.

In order to use the most complete information possible, a full additive model was constructed first (Eq. (1)). The method used for estimation of GPP (GPPmeth) was specified as a random effect on the intercept, as visual inspection suggested that CUE values were smaller where 'scaling' was used to estimate GPP compared to cases where 'micromet' was used to estimate GPP.

In Eq. (1) the variable 'age' represents the development status of the vegetation, i.e. either average age of the canopy forming trees or the period since the last major disturbance. The other three parameters represent different aspects of the climate. The absolute latitude, |lat|, was chosen as a proxy of radiation climate, i.e. day length and the seasonality of daily radiation. The term $\eta Z_{GPPMeth}$ represents the random effect on the intercept due to the different methods of estimating GPP.

These variables were not independent (Supplementary Table 1). If the different driver variables contain information that is not included in any of the other driver variables, multiple linear regression is nonetheless able to separate the individual effects. If, on the contrary, two variables exert essentially the same effect on the response variable (CUE) this can be seen in an ANOVA based model comparison. These considerations led us to the selection procedure in which we started with the full model (Eq. (1)) and compared it with all possible reduced models (Supplementary Table 2). The result of this analysis is the model with the smallest number of parameters that does not significantly differ from the full model.

We also examined, whether there were any significant interactions of predictor variables. There were not.

We used the R function lmer from the R-package lme4[63] to fit the mixed and ordinary multiple linear models to the data. We checked for potential problems of multicollinearity using the variance inflation factor (VIF)[64]. All predictors had VIF < 5 (between 1.1 and 3.8). The model residuals were also tested for normality (using

the Anderson-Darling test of non-normality, in the R-package nortest[65]. For models that did not take a random intercept regarding 'GPPmeth' into account (16–30 in Supplementary Table 2) the Anderson-Darling test found significant deviation from normality of the model residuals, hence these models were excluded from the analysis. The remaining models were compared with one another using the function ANOVA of the R-package lmerTest[66]. This resulted in a $15 \times 15$ matrix of model comparisons in which the full model turned out to be significantly different from all other models.

The same analysis was also performed with a log-transformed version of Eq. (1):

$$log(\text{FPE}) = \beta'_0 + \beta'_1 ln(\text{MAT} + 7.5) + \beta'_2 ln(\text{age}) \\ + \beta'_3 ln(\text{TAP}) + \beta'_4 ln(|\text{lat}|) + \eta' Z_{\text{GPPmeth}} + \varepsilon \quad (5)$$

where 7.5 °C was added to MAT in order to make its minimum 1 °C. Note that the linear model from the log-transformed variables differs from the untransformed linear model. The coefficients, here noted with a prime, can be interpreted on the basis of the back-transformed model. Contrary to the untransformed linear model where effects are additive, the back-transformed model is a multiplicative effect model, with the slope parameters as exponents for each variable and the intercept ($\beta'_0$ as power of $e$). As with the untransformed model, negative slope parameter values lower CUE, positive increase it with increasing driver variable values.

The results from this analysis were, as with the original additive model (Eq. (1)), (i) the full model could not be reduced any further and (ii) the directions of the effects were the same as with the additive model, i.e. the predicted CUE increased with increasing MAT, TAP and |lat| but decreased with increasing age.

The AIC and BIC values were lower for the log-transformed model compared to the untransformed model, with AIC values of −169.7 and −157.2 and BIC values of −149.0 and −136.5 for the log-transformed and untransformed models, respectively. The coefficients and model performance parameters of the untransformed and the log-transformed models are shown in Table 1 and Supplementary Table 4. The adjusted squared correlation coefficients were similar: 0.306 for the untransformed and 0.321 for the log-transformed model. Despite considerable uncertainty of the CUE values, it was possible to derive significant, systematic, linear relationships between the four driver variables and CUE or ln (CUE). Both model variants showed the same direction and similar magnitudes of the effects. It can be concluded that CUE (or ln (CUE)) from a global dataset of a large variety of forests is significantly positively affected by MAT, TAP and |lat|, and significantly negatively affected by age. Even excluding from the analysis the five tropical forest data with |lat| < 20 degrees did not alter significantly the empirical relationship (Supplementary Table 5).

Because the parameters of the untransformed, additive model are much easier to interpret, we use the additive model in the main text and use the log-transformed model only as a confirmation of trends found in the additive model.

**Outputs from TRENDY v.7**. We used the simulations from eight Dynamic Global Vegetation Models (DGVMs) performed in the framework of the TRENDY v.7 project[2,67] (http://dgvm.ceh.ac.uk/node/9; data downloaded 27 November 2019). Models that did not provide NPP and GPP at plant functional type level were excluded because of the need to analyse CUE in forests without significant contributions from shrubs, grassland or crops. The selection comprises the following models: ISAM, JULES, LPJ-GUESS, CABLE-POP, ORCHIDEE, ORCHIDE-CNP, JSBACH and SDGVM (for references on models see refs. [2,67] and Supplementary Table 6). All the models represent the surface fluxes of $CO_2$, water and the dynamics of carbon pools in response to changes in climate, atmospheric $CO_2$ concentration, and land-use change across a global grid. However, processes underlying the exchanges of water and carbon are based on different formulations in different models.

In the TRENDY protocol all DGVMs were forced with common historical climate fields and atmospheric $CO_2$ concentrations over the period from 1700 to 2017. Climate fields were taken from the CRU-JRA55 dataset[2], whereas the time series of atmospheric $CO_2$ concentrations were derived from the combination of ice core records and atmospheric observations. Land-use change was taken into account in the simulations (S3). However, similar simulations without land-use change (S2) were also tested, showing no differences. CUE was estimated as NPP/ GPP (where NPP is commonly obtained in models by subtracting $R_a$ from GPP) for the forest plant functional types simulated to be present in each grid cell. The model outputs refer to the mean from 1995 to 2015 for comparability with the records used when showing global land analysis (Fig. 5 and Supplementary Fig. 4). At site level, the same dates as the observations were chosen from the model outputs.

**Reporting summary**. Further information on research design is available in the Nature Research Reporting Summary linked to this article.

## Data availability
All data supporting this study are available in the supplementary materials and are publicly available at theZenodo repository (https://doi.org/10.5281/zenodo.3953478). Correspondence and requests for additional materials should be addressed to A.C. and A. I. Source data are provided with this paper.

## Code availability
There is no particular custom code or mathematical algorithm that is deemed central to the conclusions. All relevant R-functions that were used are referred to in the method section (see package vignettes for details).

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

# ARTICLE

27. Ryan, M. G., Binkley, D., Fownes, J. H., Giardina, C. P. & Senock, R. S. An experimental test of the causes of forest growth decline with stand age. *Ecol. Monogr.* **74**, 393–414 (2004).

28. Reich, P. B. et al. Scaling of respiration to nitrogen in leaves, stems and roots of higher land plants. *Ecol. Lett.* **11**, 793–801 (2008).

29. Mori, S. et al. Mixed-power scaling of whole-plant respiration from seedlings to giant trees. *Proc. Natl Acad. Sci. USA* **107**, 1447–1451 (2010).

30. Johnson, D. W. Progressive N limitation in forests: review and implications for long-term responses to elevated CO2. *Ecology* **87**, 64–75 (2006).

31. Gill, A. L. & Finzi, A. C. Belowground carbon flux links biogeochemical cycles and resource-use efficiency at the global scale. *Ecol. Lett.* **19**, 1419–1428 (2016).

32. Way, D. A. & Sage, R. F. Elevated growth temperatures reduce the carbon gain of black spruce [Picea mariana (Mill.) B.S.P.]. *Glob. Chang. Biol* **14**, 624–636 (2008).

33. Collalti, A. et al. Thinning can reduce losses in carbon use efficiency and carbon stocks in managed forests under warmer climate. *J. Adv. Model. Earth Syst.* **10**, 2427–2452 (2018).

34. Michaletz, S. T., Cheng, D., Kerkhoff, A. J. & Enquist, B. J. Convergence of terrestrial plant production across global climate gradients. *Nature* **512**, 39–43 (2014).

35. Malhi, Y. The productivity, metabolism and carbon cycle of tropical forest vegetation. *J. Ecol.* **100**, 65–75 (2012).

36. Merganičová, K. et al. Forest carbon allocation modelling under climate change. *Tree Physiol.* **39**, 1937–1960 (2019).

37. Sala, A. & Hoch, G. Height-related growth declines in ponderosa pine are not due to carbon limitation. *Plant. Cell Environ.* **32**, 22–30 (2009).

38. Dietze, M. C. et al. Nonstructural carbon in woody plants. *Annu. Rev. Plant Biol.* **65**, 667–687 (2014).

39. Metcalfe, D. B. et al. Shifts in plant respiration and carbon use efficiency at a large-scale drought experiment in the eastern Amazon. *N. Phytol.* **187**, 608–621 (2010).

40. Ibrom, A. et al. Variation in photosynthetic light-use efficiency in a mountainous tropical rain forest in Indonesia. *Tree Physiol.* **28**, 499–508 (2008).

41. Larcher, W. *Physiological Plant Ecology* (Springer-Verlag Berlin Heidelberg, 2003).

42. Drake, J. E. et al. Does physiological acclimation to climate warming stabilize the ratio of canopy respiration to photosynthesis? *N. Phytol.* **211**, 850–863 (2016).

43. Gifford, R. M. Plant respiration in productivity models: conceptualisation, representation and issues for global terrestrial carbon-cycle research. *Funct. Plant Biol.* **30**, 171–186 (2003).

44. O'Leary, B. M., Asao, S., Millar, A. H. & Atkin, O. K. Core principles which explain variation in respiration across biological scales. *N. Phytol.* **222**, 670–686 (2019).

45. Smith, N. G. & Dukes, J. S. Plant respiration and photosynthesis in global-scale models: incorporating acclimation to temperature and CO2. *Glob. Chang. Biol.* **19**, 45–63 (2013).

46. Wang, H. et al. Acclimation of leaf respiration consistent with optimal photosynthetic capacity. *Glob. Chang. Biol.* **26**, 2573–2583 (2020).

47. He, Y., Piao, S., Li, X., Chen, A. & Qin, D. Global patterns of vegetation carbon use efficiency and their climate drivers deduced from MODIS satellite data and process-based models. *Agric. Forest. Meteorol.* **256–257**, 150–158 (2018).

48. Griffin, K. L. & Prager, C. M. Where does the carbon go? Thermal acclimation of respiration and increased photosynthesis in trees at the temperate-boreal ecotone. *Tree Physiol.* **37**, 281–284 (2017).

49. O'sullivan, O. S. et al. Thermal limits of leaf metabolism across biomes. *Glob. Chang. Biol.* **23**, 209–223 (2017).

50. VOGEL, J. G. et al. Carbon allocation in boreal black spruce forests across regions varying in soil temperature and precipitation. *Glob. Chang. Biol.* **14**, 1503–1516 (2008).

51. Sperling, O., Earles, J. M., Secchi, F., Godfrey, J. & Zwieniecki, M. A. Frost induces respiration and accelerates carbon depletion in trees. *PLoS ONE* **10**, e0144124–e0144124 (2015).

52. Thornley, J. H. M. Plant growth and respiration re-visited: maintenance respiration defined—it is an emergent property of, not a separate process within, the system—and why the respiration: photosynthesis ratio is conservative. *Ann. Bot.* **108**, 1365–1380 (2011).

53. Landsberg, J. J., Waring, R. H. & Williams, M. Commentary on the assessment of NPP/GPP ratio. *Tree Physiol.* https://doi.org/10.1093/treephys/tpaa016 (2020).

54. Chapin, F. S. et al. Reconciling carbon-cycle concepts, terminology, and methods. *Ecosystems* **9**, 1041–1050 (2006).

55. Luyssaert, S. et al. CO2 balance of boreal, temperate, and tropical forests derived from a global database. *Glob. Chang. Biol.* **13**, 2509–2537 (2007).

56. Campioli, M. et al. Evaluating the convergence between eddy-covariance and biometric methods for assessing carbon budgets of forests. *Nat. Commun.* **7**, 13717 (2016).

57. Collalti, A. et al. Forest production efficiency increases with growth temperature—dataset. *BioRxiv* https://doi.org/10.5281/zenodo.3953478 (2020).

58. Curtis, P. S. et al. Respiratory carbon losses and the carbon-use efficiency of a northern hardwood forest, 1999–2003. *N. Phytol.* **167**, 437–456 (2005).

59. Law, B. E., Thornton, P. E., Irvine, J., Anthoni, P. M. & Van Tuyl, S. Carbon storage and fluxes in ponderosa pine forests at different developmental stages. *Glob. Chang. Biol.* **7**, 755–777 (2001).

60. Dore, S. et al. Carbon and water fluxes from ponderosa pine forests disturbed by wildfire and thinning. *Ecol. Appl.* **20**, 663–683 (2010).

61. Goulden, M. L. et al. Patterns of NPP, GPP, respiration, and NEP during boreal forest succession. *Glob. Chang. Biol.* **17**, 855–871 (2011).

62. Slob, W. Uncertainty analysis in multiplicative models. *Risk Anal.* **14**, 571–576 (1994).

63. Bates, D., Mächler, M., Bolker, B. & Walker, S. Fitting linear mixed-effects models using lme4. *J. Stat. Softw.* **67**, 1–48 (2015).

64. Kumar, K. N. R. *Econometrics* (Narendra Publishing House, 2020).

65. Gross, J. & Ligges, U. nortest: tests for normality. R package version 1.0-4. https://cran.r-project.org/web/packages/nortest/index.html (2015).

66. Kuznetsova, A., Brockhoff, P. B. & Christensen, R. H. B. lmerTest package: tests in linear mixed effects models. *J. Stat. Softw.* **82**, 1–26 (2017).

67. Sitch, S. et al. Recent trends and drivers of regional sources and sinks of carbon dioxide. *Biogeosciences* **12**, 653–679 (2015).

## Acknowledgements

We thank R.H. Waring, S. Vicca, M. Campioli, F. Pagani and E. Grieco for early constructive comments and thoughtful suggestions; S. Noce for the map of data points. We thank efforts from all site investigators and their funding agencies. This paper contributes to the AXA Chair Programme in Biosphere and Climate Impacts and the Imperial College initiative Grand Challenges in Ecosystems and the Environment. A.C. and G.M. are partially supported by resources available from the Ministry of University and Research (FOE-2019), under the project "Climate Change" (CNR DTA.AD003.474); M.F.-M. is a postdoctoral fellow of the Research Foundation—Flanders (FWO);

## Author contributions

A.C., A.I. and I.C.P. conceived the paper. A.Co., A.S., A.I., A.Ce. and R.A. analysed data. A.Co., A.I., A.Ce., R.A., M.F.-M. and I.C.P. wrote the manuscript. All authors contributed substantially to discussions and revisions.

## Competing interests

The authors declare no competing interests.
