## [Peer Review File · Nature Communications]

Reviewers' Comments:

Reviewer #1:

Remarks to the Author:

Comments on "Forest production efficiency increases with growth temperature" submitted by A. Collalti et al. to Nature Communications

General points

This is an interesting piece of work for me and I enjoyed reading the manuscript. In this study, the authors explored the global relationships of forest production efficiency (FPE: i.e. carbon use efficiency and biomass production efficiency), with biological and environmental factors. They constructed a database of observations for productivities (GPP and NPP), biomass production, and explanatory variables. By applying the mixed-effects multivariate linear regression, they found positive relationships with latitude, temperature, and precipitation, and a negative relationship with stand age. The positive relationship with temperature is, as long as I know, a novel finding in terms of both observations and process-based modelling. Then, they discussed underlying mechanisms for the counterintuitive relationship, such as thermal acclimation and nutrient acquisition cost. As demonstrated by the relationships in TRENDY outputs, the contemporary models do not capture the positive relationship between temperature and FPE, implying a serious bias in the present estimation of global carbon budget. In this sense, this study has considerable importance for improving our predictability of the global carbon budget.

There remain, however, several technical concerns in this study. I agree that the authors made great effort to compile the database (as shown by global coverage in Figure S1), but it could contain some biases. For example, Table S1 shows that MAT is significantly correlated with age. Although the authors defend the statistical approach adopted in this study (Line 125-126, Line 427-429), I suppose that the multicollinearity should have affected the results. At least, the authors should show the values of variance inflation factor (VIF) which is also available by R packages. Second, in Figure 4, I could not understand why the predicted FPE are positively correlated with both MAT and absolute latitude, because MAT is likely to be high in low latitudes.

The manuscript is well written overall and so I recommend that the manuscript is acceptable for publication after minor revision. The paper would provide effective insights for vegetation modelers and stimulate plant ecophysiological works.

Specific points

Line 82: Giving definitions of CUE, BPE, and FPE as equations (with support of Figure S2, like Eq. 3) would help readers to avoid confusion for these terms.

Line 180: Difference in herbivory intensity may explain a part of the variability in CUE and BPE.

Line 197: Can you resolve the age-gradient of FPE into age-gradients of GPP, NPP, and FP? This may help readers to understand the reason of the age-FPE relationship.

Line 265-266: "Heat tolerance ... increase linearly with ... latitude". Is it correct? Figure 2a of the paper (O'Sullivan et al. 2017, GCB) shows a negative relationship.

Line 388-403: The procedure to calculate the data uncertainty looks a bit arbitrary, although it was derived from a precedented work. Just a comment, no reply needed.

Line 444: I understand the model selection strategy, but still I recommend showing AIC and BIC values for the reduced models.

Reviewer #2:

Remarks to the Author:

In this manuscript, Collalti et al compiled a database of 244 records across >100 forest sites to look at

aboveground forest production efficiency and how it varies with climate (predominantly temperature, precipitation, and indirectly radiation through the use of latitude in their models). The authors find that FPE increases with temperature across the studies that they compiled. Further the authors find the opposite trend across 8 vegetation models included in the TRENDY project. This result goes against hypotheses included in the TRENDY models where there is increased respiration sensitivity with higher temperature. The authors thus hypothesize that the diverging trend between models and observations is due to acclimation of respiration, a process not included in the TRENDY models. This is a very interesting study and will result in a high number of citations, if for nothing else then the impressive database of FPE studies that the authors invested an enormous amount of work compiling (provided that the authors make it open access, which seems to be their intention based on my reading of the acknowledgements). Still, I have several significant reservations that need to be addressed before publication.

My biggest concern is that in the majority of the panels in Fig. 4 there are very few tropical data points and a large variability at more temperate latitudes. I'm worried that the handful of tropical data points could be driving significance trends. In particular, Fig 4d has just 2 points in blue and 3 points in green with $|\text{lat}| < 20$ degrees. If the authors exclude these points, do they still get significant climate relationships? I would like to see the sensitivity of the authors' conclusions to these data points.

Second, though well written overall, I found the organization to be difficult given the format of NC where the methods are at the end. I felt very in the dark about what was meant by 'Micromet' and 'Scaling' and it would be helpful for the authors to include this upfront in the main text. Some structural suggestions are in my line specific comments. Further, Fig. 1, Fig 2a, and Fig 3 are more methods oriented and might be more appropriate for the SI. I also found that some Figures/ Tables were mentioned out of order. Finally, some of the methods are still unclear and need work (see below comments)

A few really positive things about the MS: 1) The database that was compiled is incredible useful for many ecologists and modelers, I commend the authors and expect it to be well used. 2) As a vegetation modeler, Fig 2b is very useful. I often find papers in Nature journals citing how their results could be used in models, but few deliver on this promise. 3) Table S5 is a useful reference. 4) In Fig 5, I find it really interesting to see the sensitivity of different PFT parameterizations to MAT, particularly highlighted by LPJ Gues.

My line specific comments are listed below:

General methods questions:

The authors mention testing the temperature response while holding latitude constant in the discussion but I couldn't find this in the methods.

It's not clear why the authors didn't look at other climate variables. This could explain the significant spread within a particular latitude (fig 4). The choice of climate variables needs to be justified.

The authors need to be clear that FPE is aboveground only. Indeed, this is part of the discussion at the end where boreal root allocation is mentioned, so it should be clear at the beginning in the definition.

L52-53 'Earlier findings – FPE declining with age – are supported by this analysis.' This is a weird sentence for the abstract and it detracts from the authors work to put this first. It's a really cool result, but I wouldn't highlight it in the abstract for a general audience like NC.

L106 Fig 1 shows the variability but it does not show spatial variability. Please omit 'spatial'
Fig 2 it would be good to have different colors for panels a and b because the categories are different

Fig 2 panel and L 112-114 might be more appropriate for the SI given the location of the methods later on in the text

L114-117 Cool!

Table 1 please write out all acronyms for the fixed effects in the table legend. Also, it would be good if the authors performed a nonparametric test like Spearman's rho. Same comment for Table S1

Fig 3 can the authors include the Waring stars in the figure legend and also make these points larger so they stand out better? This figure may be more appropriate for the SI

The authors skip directly from Fig 3 to Fig 5

Fig 5 is the site-level all of the models combined? Are all the time points averaged over for the same grid cell? It would be good to show panel 4a with this. For direct comparison. Why didn't the authors also look at TAP and lat?

Fig S2 out of order

What is the time period for the data used versus model?

L235 be specific about what models (I think you mean the TRENDY ones)

L240-241 well there is potentially the Kok effect...

Paragraph 253-261 can you elaborate on the emergent constraint method so the readers don't have to go to He et al to understand this paragraph?

L315 this is aboveground biomass production, no? It would not represent boreal ecosystems allocating more to roots. The authors need to be clear as this was an important point in the discussion

L325 what does 'mostly from 1995 -2015' mean. Be specific about the entire time interval

L323 Can the authors be more specific about what these data actually are? Were they just measured over 1 year?

L323 how did the authors select these 300 peer reviewed articles?

L340 what did the authors do with the data by different authors for the same site?

L352 did the authors look at nutrient status? (they say some of the plots were N fertilized)

L368 the authors definition of terms might fit better here as it is integral in your estimation method

L374-386 these definitions in some form need to be up in the main text, I felt in the dark without them

L388-390 I was not familiar with the Luysaert method and the latitudinal dependence seems very

strange to me so I took a look at the paper. It would be nice for the authors to provide a 1-2 sentence for the latitudinal explanation so other readers can better understand why it is a useful way to estimate error to help them decide if they want to spend the time going to the Luyssaert paper

L417 Im a little confused by this because it seems like the gpp methods were separated into different models (different colors, fig 4)? Could the authors help me understand how the random effect (GPPmeth) is different from this binning of different data

Table 1: are these coefficients standardized or the raw response coefficients. Judging from the use in Fig 4 they are raw, which makes sense in that context, but it is important to be clear so the reader doesn't compare the sensitivity

L492 was CUE calculated as the average over the 20 years? It is unclear how the temporal component factors into the analysis. The statistical modeling part of the TRENDY model analysis needs more explanation

Reviewer #3:

Remarks to the Author:

Review of NCOMMS-20-12586: Forest production efficiency increases with growth temperature by Collalti et al.

Collalti et al. assemble and analyze a large global dataset to explore: (1) the range of carbon use efficiency (CUE) and biomass production efficiency (BPE) values present in forests to compare to prior published work; and (2) relationships between these metrics and stand (i.e., age) and environmental (i.e., climate) variables. The overall goal was to build upon prior studies (with much smaller data sets) to test current understanding and use of these concepts in dynamic models. In line with prior studies, they found that there is a wide range in CUE and BPE estimates but that mean values center around 0.47. Moreover, they found that these variables decline with age (also seen in prior studies), while increasing with latitude, precipitation and temperature. This last observed relationship (i.e., declining CUE and BPE with temperature is contrary to current understanding and treatment in models). Much attention, therefore, is given to this point as it is commonly not included in models as such (rather, the opposite is true of most/all current models).

Overall, I found this work to be timely, important and well presented. While the average values presented herein are almost identical to initial work done by Waring et al. some 20+ years ago, they are based on well more than 200 observations compared to the 12 used in Waring et al. (which also included many assumptions to estimate CUE that potentially constrained the results). In addition, evidence of declining CUE with stand age was first reported by DeLucia et al. 2007. However, that analysis appeared to be anchored by a single study with an unrealistically high CUE estimate (in excess of 0.8 if I remember correctly), and eliminating that single data point resulted in no relationship whatsoever between stand age and CUE. As such, the current study is a significant advance over the prior studies that were limited by much smaller datasets at the time they were published. Importantly, CUE and BPE are critically important components of the dynamic vegetation models that allow the prediction of future forest C dynamics. If all of those models get any of this wrong to start with, the results of those models are suspect at best. I have no major revision suggestions, but a series of more minor suggestions for the authors to consider. I very much enjoyed reading this article, feel that it is timely and important for the field, and will be of great interest to ecologists and modelers globally.

1) CUE vs. BPE vs. FPE. The authors go to great length to define and outline the terms CUE and BPE, including their differences, assumptions and potential problems. They then show that these two estimates were statistically indistinguishable in their data set, so move to Forest Production Efficiency (FPE) thereafter (and in the title, abstract, etc.). My suggestion is to just use FPE from the very get go, and in defining it talk about its relationship to CUE and BPE. I found the current presentation and use of the three terms to be a bit distracting considering that ultimately FPE was used for most/all analyses. It strikes me that FPE is an overarching term that includes and, in the case of this article, subsumes CUE and BPE.

2) Relationship of FPE and temperature. This is likely to be one of the most important, and potentially controversial, results of this study as it goes against current understand of plant responses to rising temperature, as the authors highlight. With that in mind, I encourage the authors to consider two points. First, the response of forests to long term temperature vs. the response of forests to short term increases in temperature are not necessarily the same thing. You have identified a response of forests that are acclimated to the temperature in which they are growing, vs. a response that may occur over much shorter time scales with contemporary climate change (this latter response is what many models are trying to predict). This is particularly important in forests that are at the high end of their optimum temperature range (e.g., tropical forests), which might well respond differently to rapid increases in temperature compared to temperate or boreal forests. I feel like this deserves more attention in the paper (e.g., couching this result in this context). Second, the relationship observed between FPE and temperature only accounts for 30% of the variation in the data. The authors stress that this was an "unexpectedly high" accounting of variance given the limitations of the data, but it still shows that 70% of that variation is unaccounted for. This also deserves more attention in my opinion for this finding to be more useful to the wider community of scientists. As it stands, the authors leave this as being explained by a higher cost for nutrient acquisition in boreal forests. There is evidence for this, but does that explain the entire result?

3) There are a lot of Methods sprinkled throughout the Results which made that section a bit hard to decipher. Suggest moving all methods to Methods, and focusing on results in Results.

4) Lines 77-79: Worth citing Clark et al. 2001 here (Clark, D. A., S. Brown, D. W. Kicklighter, J. Q. Chambers, J. R. Thomlinson, and J. Ni. 2001. Measuring net primary production in forests: concepts and field methods. *Ecological Applications* 11:356-370)? They do a nice job of estimating the impact on NPP values when ignoring these components.

5) Lines 195-207: I was surprised to not see Ryan et al. 2004 cited and discussed here (Ryan, M. G., D. Binkley, J. H. Fownes, C. P. Giardina, and R. S. Senock. 2004. An experimental test of the causes of forest growth decline with stand age. *Ecological Monographs* 74:393-414). To my knowledge, that is the only study that systematically examined each of these competing hypotheses in one model study system. They found that across the competing hypotheses, the age-related decline in NPP was a result of the decline in aboveground wood production being proportionally greater than the decline in canopy photosynthesis.

6) Environmental effects on FPE. Were TAP and MAT correlated in the dataset (typically they are)? If so, how did you handle this to tease apart real vs. potential autocorrelation effects on FPE?

7) Lines 269-271: See Vogel et al. (2008) for evidence to support this contention (Vogel, J. G., B. P. Bond-Lamberty, E. A. G. Schuur, S. T. Gower, M. C. Mack, K. E. B. O'Connell, D. W. Valentine, and R. W. Ruess. 2008. Carbon allocation in boreal black spruce forests across regions varying in soil temperature and precipitation. *Global Change Biology* 14:1503-1516).

Creighton M. Litton
University of Hawaii at Manoa

✓ **Reviewer# 1**

Comments on “Forest production efficiency increases with growth temperature” submitted by A. Collalti et al. to Nature Communications

General points

Comment 1: *This is an interesting piece of work for me and I enjoyed reading the manuscript. In this study, the authors explored the global relationships of forest production efficiency (FPE: i.e. carbon use efficiency and biomass production efficiency), with biological and environmental factors. They constructed a database of observations for productivities (GPP and NPP), biomass production, and explanatory variables. By applying the mixed-effects multivariate linear regression, they found positive relationships with latitude, temperature, and precipitation, and a negative relationship with stand age. The positive relationship with temperature is, as long as I know, a novel finding in terms of both observations and process-based modelling. Then, they discussed underlying mechanisms for the counterintuitive relationship, such as thermal acclimation and nutrient acquisition cost. As demonstrated by the relationships in TRENDY outputs, the contemporary models do not capture the positive relationship between temperature and FPE, implying a serious bias in the present estimation of global carbon budget. In this sense, this study has considerable importance for improving our predictability of the global carbon budget.*

Reply 1: We thank the reviewer for this positive assessment.

Comment 2: *There remain, however, several technical concerns in this study. I agree that the authors made great effort to compile the database (as shown by global coverage in Figure S1), but it could contain some biases. For example, Table S1 shows that MAT is significantly correlated with age. Although the authors defend the statistical approach adopted in this study (Line 125-126, Line 427-429), I suppose that the multicollinearity should have affected the results. At least, the authors should show the values of variance inflation factor (VIF) which is also available by R packages.*

Reply 2: We thank the reviewer for this comment that gives us the opportunity to clarify.

There are correlations among the explanatory variables, but they are moderate, with only the correlation between MAT and |lat| exceeding 0.7 (0.82). Following the comment, we have calculated the Variance Inflation Factors, both from the `vif()` function in the R package `car`, and as the diagonal in the inverse of the correlation matrix, following Harrell (2013), page 65, with similar results. There is no generally accepted level for the VIF, under which data are regarded as unproblematic. In Brown, Tauber and Walczak (2009), J. Ferré argues that a $VIF > 10$ indicates problems. Other scientists argue for a more cautious approach where the VIF limit is 5 (e.g. Kumar, 2020). But none of these limits indicate problems for our data, since the VIFs are between 1.1 and 3.8 (Line 493 – 494).

In general, problems with collinearity become more acute when the model contains interactions. This is not the case in our model.

Comment 3: *Second, in Figure 4, I could not understand why the predicted FPE are positively correlated with both MAT and absolute latitude, because MAT is likely to be high in low latitudes.*

Reply 3: The fitted effect parameters, i.e. the slopes of the graphs presented in Fig. 4 and Table 1, represent what happens when the explanatory variable is increased by one unit, *while all others are kept fixed* – i.e. these effects are ‘all else equal’. In this dataset we have been able to separate the effects of

MAT from latitude because (as indicated above by the VIFs) they are not very closely correlated, due to the inclusion of sites at different elevations and in more and less continental climates.

Comment 4: *The manuscript is well written overall and so I recommend that the manuscript is acceptable for publication after minor revision. The paper would provide effective insights for vegetation modelers and stimulate plant ecophysiological works.*

Reply 4: Many thanks!

Specific points

Comment 5: *Line 82: Giving definitions of CUE, BPE, and FPE as equations (with support of Figure S2, like Eq. 3) would help readers to avoid confusion for these terms.*

Reply 5: **We agree with the reviewer and included the equations explaining explicitly both CUE and BPE metrics and FPE at the very beginning of the introductory paragraph (from Line 83 to 90)**

Comment 6: *Line 180: Difference in herbivory intensity may explain a part of the variability in CUE and BPE.*

Reply 6: **Yes, probably; but in any case difficult to account for and highly variable both temporally and spatially, so it is not practically possible to analyse the effect of herbivory in this data set.**

Comment 7: *Line 197: Can you resolve the age-gradient of FPE into age-gradients of GPP, NPP, and FP? This may help readers to understand the reason of the age-FPE relationship.*

Reply 7: **This is an interesting idea that we also looked into. Applying the same regression model (eq. (1)) to GPP and FP (rather than FPE) did not yield significant parameters, because the residuals were not normally distributed. In fact, the strong advantage of using FPE as relative metric showing efficiency and not, e.g., the absolute flux rate, allows comparing a great variety of forests, differing largely in GPP and NPP. In this respect, the effects of the drivers may be hidden when looking at gross fluxes or net budgets, while they can be shown using FPE.**

Comment 8: *Line 265-266: "Heat tolerance ... increase linearly with ... latitude". Is it correct? Figure 2a of the paper (O'Sullivan et al. 2017, GCB) shows a negative relationship.*

Reply 8: **The reviewer gets right; Fig. 2 in O'Sullivan et al. (2017) shows that heat tolerance in leaves increases linearly with MAT (Fig. 2b in O'Sullivan et al. 2017) and decreases with increasing absolute latitude (Fig. 2a in O'Sullivan et al. 2017). Thus, we have now clarified and corrected the sentence in: "Heat tolerance in leaves has also been found increasing linearly with temperature and decreasing with increasing absolute latitude" (Line 285 – 286). We thank the reviewer for noticing this.**

Comment 9: *Line 388-403: The procedure to calculate the data uncertainty looks a bit arbitrary, although it was derived from a precedented work. Just a comment, no reply needed.*

Reply 9: **Yes, we confirm this (see also Reply 41)**

Comment 10: *Line 444: I understand the model selection strategy, but still I recommend showing AIC and BIC values for the reduced models.*

Reply 10: **We would like to avoid listing the AIC and BIC for the models, because they can cause confusion about the model selection procedure. The AIC can be better used when comparing non-nested models, which are difficult to evaluate with a standard p-value. However, in this case it is evident from Table S2 that all of the considered models are parametrically nested within the model in the first line. We**

therefore decided to use the p-value method, given that it is of better use for the implemented modelling approach. However, in the “Model selection” paragraph we describe that we compared our model with a log-transformed one and that we used AIC and BIC to quantify the comparison.

✓ Reviewer# 2

Comment 11: *In this manuscript, Collalti et al compiled a database of 244 records across >100 forest sites to look at aboveground forest production efficiency and how it varies with climate (predominantly temperature, precipitation, and indirectly radiation through the use of latitude in their models). The authors find that FPE increases with temperature across the studies that they compiled. Further the authors find the opposite trend across 8 vegetation models included in the TRENDY project. This result goes against hypotheses included in the TRENDY models where there is increased respiration sensitivity with higher temperature. The authors thus hypothesize that the diverging trend between models and observations is due to acclimation of respiration, a process not included in the TRENDY models.*

This is a very interesting study and will result in a high number of citations, if for nothing else then the impressive database of FPE studies that the authors invested an enormous amount of work compiling (provided that the authors make it open access, which seems to be their intention based on my reading of the acknowledgements). Still, I have several significant reservations that need to be addressed before publication.

Reply 11: We are pleased that the referee found our manuscript interesting and we do confirm our intention to make the database open access to everyone who might be interested. We will publish the data on Zenodo server (already at <https://doi.org/10.5281/zenodo.3953478>) immediately after the publication of the manuscript.

Comment 12: *My biggest concern is that in the majority of the panels in Fig. 4 there are very few tropical data points and a large variability at more temperate latitudes. I’m worried that the handful of tropical data points could be driving significance trends. In particular, Fig 4d has just 2 points in blue and 3 points in green with |lat| < 20 degrees. If the authors exclude these points, do they still get significant climate relationships? I would like to see the sensitivity of the authors’ conclusions to these data points.*

Reply 12: We performed the requested analysis and left out the five mentioned records collected on tropical forests. The table below shows that the exclusion does not considerably alter the empirical relationship that was described using the full data set with all the sites:

–with tropical sites (AD- test for normality p = 0.0978)

	Estimate	Std. Error	df	p-value	Signif
(Intercept)	0.188437	1.06E-01	23.9	0.08891	.
MAT	0.00604	2.47E-03	136.1	0.01556	*
age	-0.000379	1.16E-04	136.2	0.0013	**
TAP	0.000068	2.07E-05	136.1	0.00134	**
latitude	0.003899	1.59E-03	136.4	0.0157	*

–without tropical sites (AD- test for normality $p = 0.0587$)

	Estimate	Std. Error	df	p-value	Signif.
(Intercept)	0.158000	1.08E-01	34.5	0.155245	n.s.
MAT	0.005690	2.50E-03	130	0.02428	*
age	-0.000389	1.44E-04	130	0.007723	**
TAP	0.000086	2.46E-05	130.5	0.000666	***
latitude	0.004310	1.74E-03	130.6	0.014458	*

Hence, we conclude that the analysis is not particularly sensitive to the tropical sites, therefore confirming the robustness of our initial analyses.

Comment 13: *Second, though well written overall, I found the organization to be difficult given the format of NC where the methods are at the end. I felt very in the dark about what was meant by ‘Micromet’ and ‘Scaling’ and it would be helpful for the authors to include this upfront in the main text. Some structural suggestions are in my line specific comments.*

Reply 13: We agree with the referee’s concern that the specific structure of Nature Communications papers, with the Methods section at the end, suggests a need for some extra guidance to the reader in the Results section. We have followed the referee’s suggestion and included a brief description of the nature of ‘scaling’, ‘micrometeorological’ and ‘model’ terms (Line 119 – 121). Thanks for this suggestion which (we think) has improved the manuscript’s readability.

Comment 14: *Further, Fig. 1, Fig 2a, and Fig 3 are more methods oriented and might be more appropriate for the SI.*

Reply 14: We agree that Fig. 1b could be more ‘methods oriented’ and we have moved this figure to SI (now Fig. S2). However, Figs. 1a, 2a, and 3 describe how FPE (as both CUE and/or BPE) are (potentially) different (Fig. 1a) because of the methods used to estimate them (Fig. 2a), as driven by age (Fig. 2b), and substantially, on average, close to the Waring et al.’s regression (Fig. 3). Fig. 3 forms part of a more general picture starting from Waring et al.’s work, describing how their 12 records have increased over time and including consideration of uncertainty attached to the data. We have, therefore, retained Figs. 1a (now simply Fig. 1), 2a, and 3 in the main text. Please note that we find a small error in Fig. 2a (FPE for ‘scaling’ was 0.42 instead of 0.44) that we promptly corrected both in the figure and in the text.

Comment 15: *I also found that some Figures/ Tables were mentioned out of order. Finally, some of the methods are still unclear and need work (see below comments)*

Reply 15: We have re-ordered the Figures and Tables accordingly.

Comment 16: *A few really positive things about the MS: 1) The database that was compiled is incredible useful for many ecologists and modelers, I commend the authors and expect it to be well used. 2) As a vegetation modeler, Fig 2b is very useful. I often find papers in Nature journals citing how their results could be used in models, but few deliver on this promise. 3) Table S5 is a useful reference. 4) In Fig 5, I find it really interesting to see the sensitivity of different PFT parameterizations to MAT, particularly highlighted by LPJ Guess.*

Reply 16: We thank the referee for these positive comments.

My line specific comments are listed below:

General methods questions:

Comment 17: *The authors mention testing the temperature response while holding latitude constant in the discussion but I couldn't find this in the methods.*

Reply 17: We mention the possibility to look at selected data sets in the text, but do not show this, as the regression approach inherently uses this information to attribute the response to the individual driver variables (see Reply 3). Any attempt to show the effects with bivariate methods, binned or selected data could potentially be misleading because the number of observations is limited and random effects will be ignored.

Comment 18: *It's not clear why the authors didn't look at other climate variables. This could explain the significant spread within a particular latitude (fig 4). The choice of climate variables needs to be justified.*

Reply 18: We agree with the reviewer that in principle other climate variables could be used, however, in Line 394 – 396 we describe the reasons of our choice: *"By using only commonly available environmental drivers to analyse the spatial variability in CUE and BPE..."*. Ultimately, we used the climate variables that were commonly available to maximise the number of sites used in the mixed model analysis.

Comment 19: *The authors need to be clear that FPE is aboveground only. Indeed, this is part of the discussion at the end where boreal root allocation is mentioned, so it should be clear at the beginning in the definition.*

Reply 19: In fact, FPE (as all of other variables) are for both above- and below-ground. See for example in *"Estimation methods"* section definition for *'Biometric'*: *" ... If not otherwise stated, we assumed that the values included both above- and below-ground plant parts"* (Line 349 – 351).

Comment 20: *L52-53 'Earlier findings – FPE declining with age – are supported by this analysis.' This is a weird sentence for the abstract and it detracts from the authors work to put this first. It's a really cool result, but I wouldn't highlight it in the abstract for a general audience like NC.*

Reply 20: After a thorough discussion, we decided to maintain this sentence. The main reason lies in that the effect of age on FPE is still a much discussed issue. Some works, but with much less data, did not find any significant effect of age, while many others found FPE decreasing with age (e.g. potentially with increasing biomass). At the end, our results confirm a pervasive effect of age, therefore, we think is worth claiming earlier findings already at the beginning.

Comment 21: *L106 Fig 1 shows the variability but it does not show spatial variability. Please omit 'spatial'*

Reply 21: We agree with the reviewer. We used 'spatial' to indicate that the variability in question is not 'temporal'. But we do not make a spatial analysis (e.g. making a CUE map), and so we have followed the reviewer's suggestion and to delete 'spatial'.

Comment 22: *Fig 2 it would be good to have different colors for panels a and b because the categories are different*

Reply 22: We agree with the reviewer and changed the colours in Fig. 2b.

Comment 23: *Fig 2 panel and L 112-114 might be more appropriate for the SI given the location of the methods later on in the text*

Reply 23: See the comment above (Reply 14) regarding our decision not to move Fig. 2 to SI. About moving L112–114 to SI, we highlight that ‘*methods*’ (as a variable) is itself a predictor and it is part of the predictors in the linear mixed model, so it needs to be mentioned here.

Comment 24: *L114-117 Cool!*

Reply 24: Thanks!

Comment 25: *Table 1 please write out all acronyms for the fixed effects in the table legend. Also, it would be good if the authors performed a nonparametric test like Spearman’s rho. Same comment for Table S1*

Reply 25: We spell out all acronyms for the fixed effects in the Table 1 and Table S1 captions. We also added in the Table 1 caption that both the Spearman’s and Pearson’s correlation coefficients have practically the same value, when squared, i.e. 0.31.

Comment 26: *Fig 3 can the authors include the Waring stars in the figure legend and also make these points larger so they stand out better? This figure may be more appropriate for the SI*

Reply 26: We agree with the reviewer and included the “Waring’s stars” into the figure legend and made the points (actually the stars) larger and added a transparent whitish background. Regarding the reason why we decided to keep Fig. 3 in the main text, please see Reply 14.

Comment 26: *The authors skip directly from Fig 3 to Fig 5*

Reply 26: Thanks for noting this. We have corrected it.

Comment 27: *Fig 5 is the site-level all of the models combined? Are all the time points averaged over for the same grid cell? It would be good to show panel 4a with this. For direct comparison. Why didn’t the authors also look at TAP and lat?*

Reply 27: Yes, all the models are combined using same time as observation in order to be coherent with observed datasets. We want to focus on the novel and emergent findings. As described in the manuscript, we tried to apply the mixed model also to model simulations using model output variables driving the mixed model but the mixed model failed. A sufficiently detailed analysis on how the models differ in all these aspects would use more space than that available for the paper.

Comment 28: *Fig S2 out of order*

Reply 28: Corrected, many thanks!

Comment 29: *What is the time period for the data used versus model?*

Reply 29: This was mentioned in the “Data selection” paragraph, but we now specify it better: “*The majority of the data (~93%) cover the time-span from 1995 to 2015*” (Line 380 – 381) while in “Outputs from TRENDY v.7” section we describe as “*The model outputs refer to the mean from 1995 to 2015 for comparability with the records used when showing global land analysis (Figure 5 and Figure S4). At site level, the same dates as the observations were chosen from the model outputs.*” (Line 554 – 557). At site level, the CUE vs. MAT relationship showed similar slope also when using averaged TRENDY simulations over 1995 – 2015 (not shown in the paper).

Comment 30: *L235 be specific about what models (I think you mean the TRENDY ones)*

Reply 30: We meant in all vegetation models. Now clarified in the revised text.

Comment 31: *L240-241 well there is potentially the Kok effect...*

Reply 31: Yes, we agree the Kok effect is important at leaf scale, but much less at the whole plant and ecosystem scale. Anyway, we do not think it is useful to increase the level of detail here.

Comment 32: *Paragraph 253-261 can you elaborate on the emergent constraint method so the readers don't have to go to He et al to understand this paragraph?*

Reply 32: We have added new text to explain the method:

He et al. (2019) applied an emergent constraint (EC) method to narrow down the range of global mean values for biomass production efficiency (BPE) as calculated by the MSTMIP ensemble of ecosystem models. EC methods rely on the existence of a spread of simulated values of a non-observable quantity (in this case, global BPE), a good inter-model correlation between simulated values of this quantity and simulated values of an observable quantity (site-specific BPE), and data that provide an observational constraint (the Campioli et al., 2015 data set, which provided BPE data at a number of sites). The correlation between simulated global BPE and site-specific BPE was used to translate the probability distribution of observed site BPE into a probability distribution of global BPE that was narrower than the initial spread of simulated global BPE. The same method was applied within major biomes. This is a powerful approach but its validity depends on the models correctly representing the relationship between site-specific and global BPE.

Comment 33: *L315 this is aboveground biomass production, no? It would not represent boreal ecosystems allocating more to roots. The authors need to be clear as this was an important point in the discussion*

Reply 33: Our records comprise data of total biomass production or total net primary production, not only aboveground data (see also our comment above, and the “Estimation methods” section).

Comment 34: *L325 what does ‘mostly from 1995 -2015’ mean. Be specific about the entire time interval*

Reply 34: We have clarified this now, giving a quantitative figure in the text (Line 380 – 381; see also Reply 29)

Comment 35: *L323 Can the authors be more specific about what these data actually are? Were they just measured over 1 year?*

Reply 35: in “Data selection” (see Methods) we specify that data were obtained by screening more than 300 peer-reviewed papers, and adding, merging and extending some previously published database for both CUE and BPE. We also give a somewhat more in depth description of the total number of records (making clear whether they refer to CUE or BPE) and other variables characterizing the stand and the environment (e.g. LAI, origin of stands, management) (e.g. Line 377 – 380). In case of measurements over more than one year, we also describe whether the data were grouped or averaged (Line 390 – 394, see also comment below)

Comment 36: *L323 how did the authors select these 300 peer reviewed articles?*

Reply 36: More than 300 papers have been found in Web Of Science, Google Scholar and Scopus search engines by looking for terms including: “Carbon Use Efficiency”, “Biomass Production Efficiency”, “Carbon Balance” and including cited references in more recent papers and databases. A complete explanation of the literature review can be found in “Data selection” paragraph (Line 365).

Comment 37: *L340 what did the authors do with the data by different authors for the same site?*

Reply 37: We agree with the reviewer that this is useful information. Hence, in “Data selection” (see Methods) we clarified by adding: “*When the same reference for data was found in different papers or collected in different database, where possible, we used data from the original source. When different authors describe the same values for the same site, one single reference (and value) has been used (in principle the oldest one).*” (Line 391 – 394)

Comment 38: *L352 did the authors look at nutrient status? (they say some of the plots were N fertilized)*

Reply 38: Yes, we looked also at the nutrient status. Unfortunately, due the very low number of dataset that mentioned fertilization (our database contains only 12 records mentioned as “fertilized”) we had to omit nutrient status as a potential driving variable in the statistical analysis.

Comment 39: *L368 the authors definition of terms might fit better here as it is integral in your estimation method*

Reply 39: We agree with the suggestion and we have moved the paragraph ‘Definition of terms’ and ‘Estimation methods’ close to each other.

Comment 40: *L374-386 these definitions in some form need to be up in the main text, I felt in the dark without them*

Reply 40: We agree with the reviewer (see also comment above) and included a brief description of the ‘methods’ already in the Results section. Thanks for having highlighted this once more.

Comment 41: *L388-390 I was not familiar with the Luysaert method and the latitudinal dependence seems very strange to me so I took a look at the paper. It would be nice for the authors to provide a 1-2 sentence for the latitudinal explanation so other readers can better understand why it is a useful way to estimate error to help them decide if they want to spend the time going to the Luysaert paper*

Reply 41: To our knowledge Luysaert et al.’s method (already applied also in Vicca et al. 2012; Campioli et al. 2015, 2016, see references below) is a simple method ‘*determined by expert judgment*’ for uncertainty estimates as it can be generally and safely applied to all relevant variables (e.g. GPP and NPP or BP). The reason for increasing uncertainties with decreasing latitude lies in the nature of the relative uncertainty metric, i.e. uncertainty increases with absolute flux magnitude. For the sake of clarity we mention in the revised text: “... *uncertainty thus decreases linearly with increasing latitude for GPP and for NPP and BP, because we assumed that the uncertainty is relative to the magnitude of the flux, which also decreases with increasing $|lat|$* ”. (Line 445 – 447)

Comment 42: *L417 I’m a little confused by this because it seems like the gpp methods were separated into different models (different colors, fig 4)? Could the authors help me understand how the random effect (GPPmeth) is different from this binning of different data*

Reply 42: The mixed model approach that we used is more powerful than performing the analysis separately in bins. The random effect allows for an intercept specific to the classes of random effect variables, while it keeps the fixed effects (slopes) global to the entire data set. Randomizing an effect thus transfers the variation from the systematic part of the model to the random part, thus incorporating the variation of the variable within the uncertainty of the model. In this way, one analyses the slope (here, the sensitivity of FPE to predictor variables) while correcting for small differences in FPE values across different classes of the random effect. In a binned analysis, by contrast, the slopes would be specific to every bin and the degrees of freedom would be higher. It also means that when using the

regression model to make predictions, you need not specify a bin. (The price is a higher uncertainty, as both the uncertainty of bins and the residual variation need to be taken into account).

Comment 43: *Table 1: are these coefficients standardized or the raw response coefficients. Judging from the use in Fig 4 they are raw, which makes sense in that context, but it is important to be clear so the reader doesn't compare the sensitivity*

Reply 43: **We mentioned in the text that the sensitivities are as calculated from the regression approach and that their values and units cannot be directly compared.**

Comment 44: *L492 was CUE calculated as the average over the 20 years? It is unclear how the temporal component factors into the analysis. The statistical modeling part of the TRENDY model analysis needs more explanation*

Reply 44: **At site level, modelled CUE from TRENDY simulations was extracted based on same dates as the observations, except for sites with unknown dates, where the modelled CUE values are the average values over the 20 years. The average values over the 20 years are also used when showing global land CUE (Fig. 5 and S4). This is now better specified in the revised version (Line 554 – 557).**

✓ **Reviewer #3**

Comment 45: *Review of NCOMMS-20-12586: Forest production efficiency increases with growth temperature by Collalti et al.*

Collalti et al. assemble and analyze a large global dataset to explore: (1) the range of carbon use efficiency (CUE) and biomass production efficiency (BPE) values present in forests to compare to prior published work; and (2) relationships between these metrics and stand (i.e., age) and environmental (i.e., climate) variables. The overall goal was to build upon prior studies (with much smaller data sets) to test current understanding and use of these concepts in dynamic models. In line with prior studies, they found that there is a wide range in CUE and BPE estimates but that mean values center around 0.47. Moreover, they found that these variables decline with age (also seen in prior studies), while increasing with latitude, precipitation and temperature. This last observed relationship (i.e., declining CUE and BPE with temperature is contrary to current understanding and treatment in models). Much attention, therefore, is given to this point as it is commonly not included in models as such (rather, the opposite is true of most/all current models).

Overall, I found this work to be timely, important and well presented. While the average values presented herein are almost identical to initial work done by Waring et al. some 20+years ago, they are based on well more than 200 observations compared to the 12 used in Waring et al. (which also included many assumptions to estimate CUE that potentially constrained the results). In addition, evidence of declining CUE with stand age was first reported by DeLucia et al. 2007. However, that analysis appeared to be anchored by a single study with an unrealistically high CUE estimate (in excess of 0.8 if I remember correctly), and eliminating that single data point resulted in no relationship whatsoever between stand age and CUE. As such, the current study is a significant advance over the prior studies that were limited by much smaller datasets at the time they were published. Importantly, CUE and BPE are critically important components of the dynamic vegetation models that allow the prediction of future forest C dynamics. If all of those models get any of this wrong to start with, the results of those models are suspect at best. I have no major revision suggestions, but a series of more minor suggestions for the authors to consider. I very much enjoyed reading this article, feel that it is timely and important for the field, and will be of great interest to ecologists and modelers globally.

Reply 45: We are grateful for the positive assessment by Prof. Litton and have used his comments to be more specific on the still rather vague knowledge of age relationships prior to our study. We hope that this new revised version has also clarified the minor concerns.

Comment 46: 1) CUE vs. BPE vs. FPE. The authors go to great length to define and outline the terms CUE and BPE, including their differences, assumptions and potential problems. They then show that these two estimates were statistically indistinguishable in their data set, so move to Forest Production Efficiency (FPE) thereafter (and in the title, abstract, etc.). My suggestion is to just use FPE from the very get go, and in defining it talk about its relationship to CUE and BPE. I found the current presentation and use of the three terms to be a bit distracting considering that ultimately FPE was used for most/all analyses. It strikes me that FPE is an overarching term that includes and, in the case of this article, subsumes CUE and BPE.

Reply 46: We agree with the reviewer and, accordingly with his suggestion, we face directly the issue of multiple variables by describing at the very beginning of the introduction with: *“Therefore we assessed estimates of both BPE and CUE as a single metric, hereafter called ‘forest production efficiency’ (FPE) and making distinctions between them when possible”* (Line 88 – 90).

Comment 47: 2) Relationship of FPE and temperature. This is likely to be one of the most important, and potentially controversial, results of this study as it goes against current understand of plant responses to rising temperature, as the authors highlight. With that in mind, I encourage the authors to consider two points. First, the response of forests to long term temperature vs. the response of forests to short term increases in temperature are not necessarily the same thing. You have identified a response of forests that are acclimated to the temperature in which they are growing, vs. a response that may occur over much shorter time scales with contemporary climate change (this latter response is what many models are trying to predict). This is particularly important in forests that are at the high end of their optimum temperature range (e.g., tropical forests), which might well respond differently to rapid increases in temperature compared to temperate or boreal forests. I feel like this deserves more attention in the paper (e.g., couching this result in this context).

Reply 47: We thank Prof. Litton for highlighting this important point. We completely agree that short- vs. long-term responses to temperature are different things. In the manuscript we specifically clarify this by writing: *“The observed increase in FPE with MAT is new, and is opposite to what would be expected based on the instantaneous responses of photosynthesis and plant respiration as described in textbooks and assumed in many process-based models.”* (Line 250 – 252) and *“However, the instantaneous response of autotrophic respiration rate is largely irrelevant here because of the longer time scale”* (Line 255 – 256) and further *“This acclimation takes place on a time scale of days to weeks. Genetic adaptation throughout multiple generations is expected to proceed in the same direction (for definitions and distinctions between acclimation and adaptation see ref. 43).”* (Line 259 – 261). Our results and our findings clearly do not refer to short-term temperature responses (as they are based on annual values, or, in some cases, even mean values over different years) but rather they represent much longer-term responses at different temperature. We have now better clarified (Line 285 – 286): *“Heat tolerance in leaves has also been found to increase linearly with temperature and to decrease with absolute latitude⁴⁹.”* Moreover, we discuss the reasons why, in our opinion, TRENDY models (but potentially many other different vegetation models not necessarily included in the TRENDY project) may fail in accounting for the temperature response, through: *“we note that the standard approach in today’s land ecosystem models as shown here, or more generally in vegetation models – where maintenance respiration per unit of respiring tissue is typically determined as a fixed basal rate at a standard temperature (commonly 15 or 20 C°), increasing with the substrate and temperature according to a fixed Q₁₀ factor or Arrhenius-type equation – cannot generate the observed positive response of CUE or BPE to growth temperature*

observed in our study." (Line 308 – 313). We agree that our work gives new insights on the potentially different effects that the expected climate change may have on the long-term on forests at the "extremes" of the temperature range (tropical and boreal forests).

Comment 48: *Second, the relationship observed between FPE and temperature only accounts for 30% of the variation in the data. The authors stress that this was an "unexpectedly high" accounting of variance given the limitations of the data, but it still shows that 70% of that variation is unaccounted for. This also deserves more attention in my opinion for this finding to be more useful to the wider community of scientists. As it stands, the authors leave this as being explained by a higher cost for nutrient acquisition in boreal forests. There is evidence for this, but does that explain the entire result?*

Reply 48: As we write, it was not expected to be able to explain an even higher fraction of inter site variability in FPE. One can argue that our model considered only a globally common set of interacting effects on FPE, while the residual variance can be possibly attributed to similar global factors, where we lack driver data for a sufficiently large number of data records, and to local factors, determined by the specific situation at the site (e.g., management, specific eco-physiology of the species involved etc.) and, of course, to random variability. At this stage, any further analysis without further information would be speculation. Hence, we confine our analysis to the novel and scientifically relevant results, i.e. the statistically significant global response pattern and the possible underlying mechanisms that constrain it.

Comment 49: *3) There are a lot of Methods sprinkled throughout the Results which made that section a bit hard to decipher. Suggest moving all methods to Methods, and focusing on results in Results.*

Reply 49: We agree with the Prof Litton's impression and we choose to amend the text with some basic information on the methods to help the reader understand the matter without the necessity to visit the method section that is generally located after the results section the end in Nat. Comm. papers. Such a guidance on methods in the main text was also suggested by another reviewer (e.g. see comment and Reply 40). Additionally, some of the variability in measured FPE is driven by succinct effects from the "methods" used (in the sense of GPP methods used to obtain the results). We have, therefore, not completely followed the advice of the referee, but if the editors think that we should remove these short methodological clarifications throughout the results sections we will be happy to do so.

Comment 50: *4) Lines 77-79: Worth citing Clark et al. 2001 here (Clark, D. A., S. Brown, D. W. Kicklighter, J. Q. Chambers, J. R. Thomlinson, and J. Ni. 2001. Measuring net primary production in forests: concepts and field methods. Ecological Applications 11:356-370)? They do a nice job of estimating the impact on NPP values when ignoring these components.*

Reply 50: We agree with the reviewer's suggestion and have added a reference to Clark et al. (2001).

Comment 51: *5) Lines 195-207: I was surprised to not see Ryan et al. 2004 cited and discussed here (Ryan, M. G., D. Binkley, J. H. Fownes, C. P. Giardina, and R. S. Senock. 2004. An experimental test of the causes of forest growth decline with stand age. Ecological Monographs 74:393-414). To my knowledge, that is the only study that systematically examined each of these competing hypotheses in one model study system. They found that across the competing hypotheses, the age-related decline in NPP was a result of the decline in aboveground wood production being proportionally greater than the decline in canopy photosynthesis.*

Reply 51: We agree with the reviewer's suggestion and have added a reference to Ryan et al. (2004).

Comment 52: *6) Environmental effects on FPE. Were TAP and MAT correlated in the dataset (typically they are)? If so, how did you handle this to tease apart real vs. potential autocorrelation effects on FPE?*

Reply 52: Collinearity of the driver variables and how the mixed model copes with it, was also asked by other referees and it was discussed at the example of MAT and |lat| above (see Reply 2 and 3). In that case, the effects on FPE from MAT and |lat| compensated each other (both positively correlated with FPE but negatively correlated with each other) while the effects from MAT and TAP support each other. This is what the model finds. The fact that the mixed model provided significant parameter values is a result from the specific information content in the data base. In others words, if the FPE response could equally well be explained by a change in MAT as, alternatively, with TAP, the parameter values were not significant.

Comment 53: 7) Lines 269-271: See Vogel et al. (2008) for evidence to support this contention (Vogel, J. G., B. P. Bond-Lamberty, E. A. G. Schuur, S. T. Gower, M. C. Mack, K. E. B. O'Connell, D. W. Valentine, and R. W. Ruess. 2008. Carbon allocation in boreal black spruce forests across regions varying in soil temperature and precipitation. *Global Change Biology* 14:1503-1516).

Reply 53: We agree and have also added Vogel et al. (2008).

.....

Yours sincerely,

Andreas Ibrom, on behalf of all authors

References (some already added to the main text):

- Brown, Tauber and Walczak (eds) (2009): *Comprehensive Chemometrics*. Elsevier 2009.
- Campioli, M. et al. Biomass production efficiency controlled by management in temperate and boreal ecosystems. *Nat. Geosci.* 8: 843–846 (2015).
- Campioli, M. et al. Evaluating the convergence between eddy-covariance and biometric methods for assessing carbon budgets of forests. *Nat. Commun.* 7: 13717 (2016).
- Harrel F (2013): *Regression Modeling Strategies*. Springer.
- Kumar NR (2020): *Econometrics*. CRC Press.
- Vicca, S. et al. Fertile forests produce biomass more efficiently. *Ecol. Lett.* 15: 520–526 (2012).

Reviewers' Comments:

Reviewer #1:

Remarks to the Author:

Comments on "Forest production efficiency increases with growth temperature" submitted by A. Collalti et al. to Nature Communications

I studied the authors' response (not only to my comments but also to other two referees) and confirmed that they made adequate revisions. Additional examinations on model's VIF and sensitivity to tropical data removal improved the scientific quality of presentation. I agree that this study would provide a useful constraint and implication to vegetation model researchers.

Reviewer #3:

Remarks to the Author:

I appreciate the author's responses to my and the other reviewers earlier comments on this manuscript. As all reviewers pointed out, this submission by Collalti et al. is timely, important, and will be of great interest to ecologists and modelers globally. It is my opinion that the authors have done a commendable job dealing with the feedback from all reviewers. There are a couple of places where the added/new text could be tightened up a bit (e.g., Lines 100-101), and some of the issues brought up by reviewers (and addressed by the authors) are "unsolvable" but far from fatal flaws. In general this is a very well written, novel and somewhat provocative (in a good way) study that will get a fair bit of exposure and interest in the carbon cycling community. Thanks for the opportunity to read and provide feedback on this work.